

# An improved glacier parameterisation for the ecLand land-surface model: local, regional and global impact

Gabriele Arduini[1], Christoph Rüdiger[2], and Gianpaolo Balsamo[3]

[1]European Centre for Medium-Range Weather Forecasts, Shinfield Park, Reading RG2 9AX, UK
[2]European Centre for Medium-Range Weather Forecasts, Robert-Schuman-Platz 3, 53175 Bonn, Germany
[3]World Meteorological Organisation, Avenue de la Paix 7bis, 1202, Genève, Switzerland

**Correspondence:** Gabriele Arduini (gabriele.arduini@ecmwf.int)

**Abstract.** Glaciers and ice sheets are critical components of the cryosphere and the climate system. In a warming climate, surface temperatures exceed the melting point more frequently and for longer periods, making ice and snowpacks increasingly susceptible to melting. Meltwater from glaciers contributes to freshwater inputs to oceans and rivers, while the ice and snow surfaces provide a cooling effect on the atmosphere. This study presents a new parameterisation enabling a more realistic representation of glaciers and ice sheets in a global land-surface model used for numerical weather prediction and reanalyses, accounting for the seasonal evolution of the snowpack and the fractional glacier coverage within grid points. The new scheme has been tested in stand-alone (offline) mode across various scales, from point-level to regional and global simulations, and validated against in situ observations and a range of reference datasets. Results show improvements in the representation of surface temperature, albedo, and snow processes compared to the current scheme, leading to a more accurate simulation of melting events and surface mass balance for the Greenland ice sheet. The impact of the new scheme on the hydrological cycle was also assessed for glacier-fed river basins, for which the increased melting generally leads to higher river discharge. It is shown that it improves simulations for basins with an underestimation of their streamflow generation during summer or early spring. However, this effect can be detrimental in basins where the current model already overestimates the discharge, further amplifying the positive bias. Overall, the new scheme enables a more accurate and physically realistic representation of glacier processes, enhancing the representation of cryosphere surfaces of future climate reanalyses for a wide range of scientific applications.

## 1 Introduction

The cryosphere is a key part of the Earth's climate system, playing a crucial role in the energy and water balance of the earth system (Serreze et al., 2007), and consequently influencing weather and climate. Glaciers and ice sheets, are two of the components of the cryosphere that are rapidly responding to climate change, reducing in size and loosing mass at an increasing rate in recent decades (Slater et al., 2021; Hock and Huss, 2021), directly impacting fresh-water input into the oceans and consequently sea level rise (Aschwanden et al., 2019; Hofer et al., 2020; Box et al., 2022). At regional scales, mountain glaciers act as water reservoirs releasing meltwater downstream during the spring and summer months (Immerzeel et al., 2020). In addition to their hydrological impact, glaciers influence regional near-surface atmospheric dynamics by creating temperature



gradients between glaciated and non-glaciated surfaces. These gradients drive local thermally-driven winds, which can impact precipitation patterns in the surroundings (Lin et al., 2021; Salerno et al., 2023). Considering large-scale impacts, Bai et al. (2025) have shown using a regional climate model for the Karakoram range in the Himalayas that mountain glaciers can influence atmospheric circulation at the synoptic scale, modifying the summer monsoon and leading to regional precipitation anomalies.

The mass balance of glaciers is driven by the accumulation and ablation of snow and ice, which in turn are driven by the energy and water fluxes at the surface, and by the movement of the ice itself at the grounding line (Khan et al., 2015; Otosaka et al., 2023). In the recent decades the surface processes, defining the surface mass balance (SMB) of the glacier, have steadily increased their contribution to the total mass loss of glaciers (Lenaerts et al., 2019; Ryan et al., 2019; IMBIE, 2020). For the Greenland ice sheet this has increased from 42% in 2000-2005 to 68% in 2009-2012, due to an increase in surface melting

and runoff (Enderlin et al., 2014). Surface processes include precipitation (solid and liquid), runoff, sublimation (which can be enhanced by wind blowing) and evaporation. The total balance can be estimated using observations, which however can be sparse (if in situ) and more importantly may require a prior knowledge or an estimate of the snow/firn density (Lenaerts et al., 2019). As an alternative, regional climate models targeting specific polar/ice sheet processes are used to estimate the surface mass balance of glaciers (see for instance Fettweis et al., 2017; Noël et al., 2018; Van Wessem et al., 2018). The advantage

of this approach is that all components of the surface mass balance are estimated consistently and available without spatial gaps. However, model outputs need to be validated against observations as uncertainties in the model physics and parameters, resolution and forcing data can introduce biases in the modelled SMB (Hermann et al., 2018).

The ice discharge at the grounding line, the latter being the boundary where ice shifts from resting on bedrock to floating on the ocean, is usually modelled using ice sheet models which account for the ice dynamics and rheology (Khan et al., 2015, for

a review). These models can be coupled to Earth System Models (ESMs) for climate applications to account for the feedbacks between the ice sheets, the atmosphere and sea-level (Nowicki et al., 2020). The time-scales involved in the dynamics of glaciers are different from those of the atmosphere, and the coupling between the two components may require a specific coupling methodology (Helsen et al., 2013). However, the usage of such models is particularly challenging for ESMs designed for Numerical Weather Prediction (NWP) applications, where initial conditions are most important. Moreover, in a seamless

modelling approach (*one-model-fits-all*), the time-scales covered by such models can range from hours (short-range forecasts) to monthly (seasonal forecasts). Therefore it is crucial to correctly initialise all model components without spurious spinups. Another approach is to parameterise the ice dynamics using a simplified model which accounts for the ice flow over a given time interval, e.g. yearly, assuming an empirical relationship between glacier thickness changes and glacier surface elevation (see for instance Huss et al., 2010; Avanzi et al., 2022).

Several land-surface models have implemented parameterisations to account for the surface mass balance of glaciated regions. Shannon et al. (2019) implemented glaciated surfaces that can exist at multiple elevations within the Joint UK Land Environment Simulator model (JULES, Best et al., 2011), to estimate the mean ice loss from glaciers in a climate change scenario. Van Kampenhout et al. (2017) introduced several changes in the Community Land Model (CLM, Lawrence et al., 2019) to improve the representation of snow over polar ice sheets. Among them, wind-driven compaction was included, and



the maximum snow mass allowed on the ice sheet was increased from 1 m to 10 m of snow water equivalent (SWE), a more realistic value for snow and firn over Greenland and Antarctica. These changes were shown to improve the snow density, as well as a better simulation of phase changes and temperature in the deep snowpack. CLM also includes elevation dependent bands to compute the surface mass balance of glaciers and can be coupled to the ice sheet model CISM (Lipscomb et al., 2013, 2019) for a full treatment of the glacier mass balance in stand-alone or fully coupled simulations.

Other regional Earth System models have included more detailed treatment of glaciers in the recent years. Mottram et al. (2017) improved the representation of melting events in the HARMONIE-AROME regional model for NWP applications, by including a physical capping to the surface temperature calculation over glaciers, using the remaining energy to melt the snowpack. This was found to reduce the surface and 2-metre temperature biases compared to in situ observations. Melting and refreezing processes are important to correctly account for the surface mass balance of glaciers, in particular as ablation period and area have been extending in recent decades (de la Peña et al., 2015; Polashenski et al., 2014). The spatio-temporal variability of snow properties was shown to be an important factor for simulating the surface mass balance of the Greenland ice sheet, in particular the handling of meltwater percolation and refreezing processes within the snowpack and the albedo parameterisation (Langen et al., 2015, 2017).

In this work, we present a new land-ice parameterisation and an update to snow processes relevant to glacier regions for the ecLand land surface model (Boussetta et al., 2021), developed at the European Centre for Medium-Range Weather Forecasts (ECMWF) and part of the Integrated Forecasting System (IFS, ECMWF, 2022). The new scheme allows for a more realistic representation of glaciers within ecLand, accounting for the seasonal evolution of the snowpack and the fractional coverage of glaciers on the grid points. Yet, it aims for simplicity and computational efficiency, to be used in global coupled simulations within the IFS for operational usage. The new scheme has been tested in stand-alone mode (i.e. land surface only simulations) at different scales, from point-scale simulations, to regional and global simulations, and compared to in situ observations and satellite products. This is made possible by the high flexibility of ecLand, which can be run in stand-alone mode, over a single point or a region, or fully coupled to the global atmospheric model.

## 2 Methodology

### 2.1 ecLand model

ecLand (Boussetta et al., 2021) is the land surface model part of the Integrated Forecasting System (IFS) developed at ECMWF, used for NWPs from daily to seasonal time-scales and reanalyses. The model is physically-based, describing the energy and water movement into the soil column and the exchange of fluxes with the atmosphere, and it can be run in a fully coupled or a stand-alone mode. While the former allows for an interactive exchange of energy and water fluxes between the land and the atmosphere, the latter is forced at the interface with external atmospheric variables without accounting for feedbacks. ecLand includes CaMa-Flood (Yamazaki et al., 2011), a river routing model which allows the simulation of river streamflow by routing the surface and subsurface runoff from the land surface model through a global river network. A full description of ecLand





is reported in Boussetta et al. (2021) and therefore in the present work only those parts relevant to this study are described in detail.

In CY49R1 (the NWP model version operational since November 2024), glaciers are parameterised in a simplistic way, by describing them as grid points entirely covered with 10 m of snow water equivalent (SWE=$10,000$ kg m$^{-2}$). Without a proper representation of sub-grid land-ice coverage, a dominant fraction approach is used to identify the grid points to be treated as glaciers, using a glacier mask indicating the fraction of a grid cell ($f_{\mathrm{gl}}$) that is covered by a glacier to create a binary glacier information. Setting $f_{\mathrm{cr}} = 0.5$ as a threshold values for dominant glacier coverage in a grid-box, grid points with $f_{\mathrm{gl}} \geq f_{\mathrm{cr}}$ are assigned a fixed SWE=$10,000$ kg m$^{-2}$, and those with $f_{\mathrm{gl}} < f_{\mathrm{cr}}$ are represented as "glacier-free points" where seasonal snow can accumulate/melt. This implies that the snow mass balances is not calculated over glacier points and that liquid water cannot be present at these grid points; in addition to that, fractional glaciers (e.g. land-ice partly covering a grid-box) are not represented in the model.

This simplified approach does not allow to simulate the seasonal variability of the snowpack over glaciers and therefore the variations in albedo and energy and water fluxes over those points for different seasons or for different atmospheric conditions. The binary glacier mask based on a threshold value also presents a challenge for changes in horizontal resolution of the model. As horizontal resolution increases, the condition $f_{\mathrm{gl}} \geq f_{\mathrm{cr}}$ can be satisfied on more grid points, and so the number of glaciers represented by the model is likely to increase. These limitations call for an updated glacier parameterisation allowing for a seasonal evolution of the snowpack and the representation of fractional glaciers.

## 2.2 New glacier parameterisation

### 2.2.1 Land-ice

In the parameterisation developed in this work, glaciers are represented using the ice tile of ecLand. In CY49R1, the ice tile was used exclusively for sea ice; in the present study, this tile is also used for grid points covered with land-ice. This allows to handle grid-point fractions covered by glaciers continuously between 0 (no glaciers) and 1 (grid-point fully covered by land-ice). This approach can be used because ecLand employs a binary land-sea mask, so that ocean and sea-ice fractions cannot coexist on the same grid-box with other land-related fractions, e.g. land-ice. A more general approach would require a separate tile for the glaciers, to allow a continuous coverage across the different media (ocean, sea-ice, land, land-ice).

Land ice is represented by 4 vertical layers of fixed depth on top of the soil column; the heat transfer within the ice is represented using the heat conduction equation:

$$(\rho C)_{\mathrm{I}} \frac{\partial T_{\mathrm{I}}}{\partial t} = \frac{\partial}{\partial z}\left[\lambda_{\mathrm{I}} \frac{\partial T_{\mathrm{I}}}{\partial z}\right], \tag{1}$$

where $(\rho C)_{\mathrm{I}} = 1.88 \times 10^6$ J m$^{-3}$ K$^{-1}$ is the volumetric ice heat capacity, $T_{\mathrm{I}}$ is the ice temperature, $\lambda_{\mathrm{I}} = 2.03$ W m$^{-1}$ K$^{-1}$ and is the ice thermal conductivity. In the absence of snow, the flux boundary condition at the top of the ice is the conductive



flux from the surface energy balance for the ice tile; at the bottom, the soil (ground) temperature is used as boundary condition and to compute the ice basal heat flux.

The albedo of land-ice is fixed to $0.4$ following Davaze et al. (2018) and Avanzi et al. (2022), to account for the presence of impurities and debris on the ice. If ice is exposed, e.g the snow is completely melted, ice can melt if the temperature of the ice is above the melting point. The amount of ice that can melt is diagnosed from the available energy for melting. The runoff generated from the ice melt is added to the surface runoff over the grid-box, assuming that the water does not infiltrate into the ice column. The change in the mass (thickness) of the ice-column is not considered in the model, as this would require a more comprehensive treatment of the ice dynamics as well, which is beyond the scope of the current work. This simplification allows considering the contribution of glaciers to runoff, without the complexity of computing the full mass balance of glaciers. Moreover, for coupled land-atmosphere simulations for NWP applications, a mass balance of the ice column would require appropriate initial conditions and spin-up for the ice prognostics variables, which is not currently feasible in operational settings.

The surface energy balance calculation, performed to diagnose the surface temperature and the surface fluxes at the new time-step using an implicit solver, is modified for the land-ice tile following the approach used in Arduini et al. (2019) for the snow tile. In case of the diagnosed surface temperature over land-ice being above the melting point, the surface energy balance is solved a second time by setting the coupling skin conductivity of the ice tile to a large number (i.e. $\lambda_{\mathrm{sk}} = 10^6 \ \mathrm{W \ m^{-2} \ K^{-1}}$) and the subsurface temperature to the melting point (see Arduini et al., 2019, for additional details). This enables the computation of a physically realistic surface temperature as well as improving the amount of energy available for melting the ice.

### 2.2.2 Modifications to the snow scheme

ecLand features an intermediate complexity multi-layer snow scheme, introduced in CY48R1 (Arduini et al., 2019). As mentioned in Sect. 2.1, the snow mass in CY49R1 was fixed to 10 m of SWE over glacier grid points (as defined in Sect. 2.2.1), with the snow density set to $300 \ \mathrm{kg \ m^{-3}}$ and the snow albedo fixed to $0.82$. In the present study, additional modifications to the snow scheme have been implemented to allow for a more realistic representation of the snowpack over glaciated surfaces.

With the introduction of the land-ice tile and fractional ice cover within a grid-box, snow can accumulate and melt on top of the ice column. A single snowpack "category" is represented within ecLand: this means that a single prognostic variable is used for each snow variable. As a consequence, for grid-points partially covered with land-ice, the grid-box average basal heat flux between the snow and the underlying surface is the weighted average of the soil and ice temperatures top layers. This can occur on grid points partially covered by land-ice, where the snowpack can be present on top of the ice and the land. The maximum snow mass allowed is set to 10 m of SWE, a value that is consistent with the one used in other land-surface models and it is a reasonable compromise between maintaining a physical realism while avoiding unrealistic accumulations of snow when the model runs over long period of times (as reported by Lawrence et al., 2019). Snowmelt can occur over glaciers when the temperature of the snowpack is above the melting point. The liquid water generated can either refreeze within the snow layer or infiltrate to the following one, using a bucket-type approach.



With the new scheme, the snow density is allowed to vary over glaciers, similar to Arduini et al. (2019). However, there are differences compared to the treatment of snow over land. The top layer of the snowpack, that is the one in contact with the atmosphere, can vary between $\rho_{\text{min,top}} = 280$ kg m$^{-3}$ (for freshly fallen snow) and $\rho_{\text{max,top}} = 315$ kg m$^{-3}$, following observations reported in Fausto et al. (2018) and modelling results from Alexander et al. (2019). The maximum snow density for the deeper layers is set at $\rho_{\text{max}} = 500$ kg m$^{-3}$, as in Arduini et al. (2019). This implies that a realistic firn-to-ice conversion,

which would require the dynamic range of snow density to vary between the one for fresh snow to pure ice, cannot be represented with the current scheme and will be considered in future work.

   In the new scheme, the snow albedo is allowed to vary over glacier points, with a revised formulation compared to the one used for land snow (Arduini et al., 2019). To take into account grid-points partially covered with land ice a weighted average between snow over land and snow over land-ice is performed:

$$\alpha_{\text{sn}} = f_{\text{gl}}\,\alpha_{\text{sn,gl}} + (1 - f_{\text{gl}})\,\alpha_{\text{sn,l}}, \tag{2}$$

with $\alpha_{\text{sn,l}}$ the snow albedo over land and $\alpha_{\text{sn,gl}}$ the snow albedo over glaciers. $\alpha_{\text{sn,l}}$ is computed as in Arduini et al. (2019), whereas $\alpha_{\text{sn,gl}}$ is computed as follows:

$$\alpha_{\text{sn,gl}} = \alpha_{\text{max,gl}} + (\alpha_{\text{sn}}^{t-1} - \alpha_{\text{min,gl}})e^{-\tau_{\text{sn,gl}}\Delta t / \tau_{\text{day}}} + \alpha_{\text{min,gl}} \qquad \text{for } T_{sn} > T_{\text{gl}} \quad, \tag{3}$$

with $\alpha_{\text{max,gl}} = 0.82$, $\alpha_{\text{min,gl}} = 0.65$ and parameters $\tau_{\text{sn,gl}} = 0.085$ s and $\tau_{\text{day}} = 86400$ s; $\alpha_{\text{sn}}^{t-1}$ is the albedo value at the

previous time-step and the temperature threshold for the albedo change is set to $T_{\text{gl}} = -5$ °C. This simple formulation allows snow albedo to vary in near-melting conditions to reflect the changes in the snowpack properties and the possible presence of liquid water on the snow surface.

## 2.3   Observation datasets

   The observational data from the Programme for Monitoring of the Greenland Ice Sheet (PROMICE, Fausto et al., 2021) include

a comprehensive and high-quality observations of surface and subsurface quantities for monitoring the climate of the Greenland ice sheet, collected from an extensive network of automated weather stations and GPS stations, and are used in this study for in situ evaluation.

   In this study only a subset of the available sites are used, displayed in Fig. 1. For evaluation purposes, sites are grouped in three categories, depending on their location and elevation: *lower* sites, *upper* sites, and *accumulation* sites, following the

classification by Fausto et al. (2021). The *lower* group includes: KAN-M, KPC-L, UPE-L, QAS-L, NUK-K, TAS-L, THU-L; the *upper* group includes: QAS-M, KPC-U, NUK-U, SCO-U, TAS-A; the *accumulation* group includes: EGP-D, CEN-D, KAN-U (see Fig. 1 for the location of the sites).

   For the regional verification over the Greenland ice sheet, the MEaSUREs Greenland Surface Melt Daily 25 km EASE-Grid 2.0 data set (Mote, 2014) is used. MEaSUREs provides daily melt/no-melt classification at 25 km resolution over the

Greenland ice sheet and it is used to evaluate the modelled melt extent and timing at the regional scale. In addition to that, the satellite derived MODIS MOD10A1 (Hall et al., 1995) albedo product, further processed by Box et al. (2017) is used. This





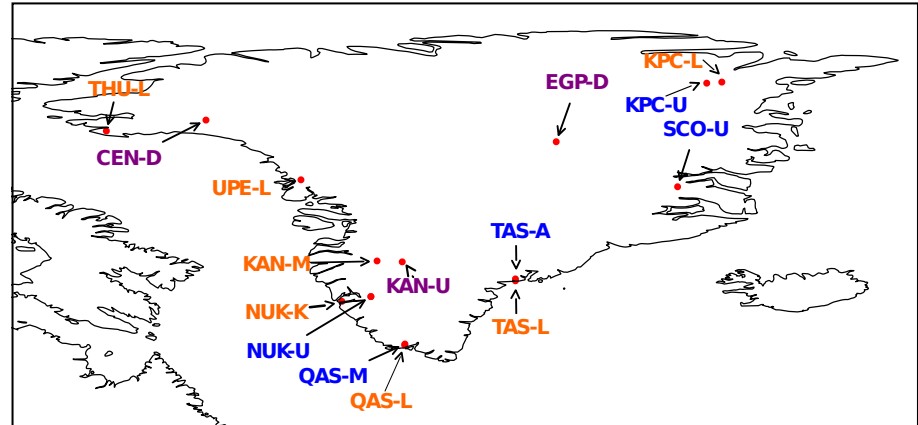

**Figure 1.** Map of the PROMICE stations used in this study. *Lower* sites are indicated in orange, *upper* sites in blue, and *accumulation* sites in purple. See Sect. 2.3 for details.

product consists of calibrated and gap-filled MODIS data specifically for the Greenland ice sheet, using the PROMICE in situ observation. The original resolution of the product is 5 km, and in this study it has been regridded to 0.15 degree to compare with the model output.

For the assessment of the surface mass balance (SMB), the SMB product for the Greenland ice sheet, described in Otosaka et al. (2022), is used. This product is based on a multi-model of Regional Climate Model (RCM) simulations and it showed a generally good agreement with other SMB estimates (Otosaka et al., 2022). However, given that this product is based on RCMs, it will be used as a reference for comparison with current state-of-the-art models for SMB studies, rather than a validation dataset.

For hydrological evaluation, a merged dataset is used, which consists of the Global Runoff Data Centre (GRDC) dataset supplemented by data from the CARAVAN dataset (Kratzert et al., 2023) and the CARAVAN extension for Iceland from Helgason and Nijssen (2024). Streamflow observations for river basins contributed by glaciers located in the Himalya regions are not included in the available observation datasets. This is an important region to be included in the evaluation, as glacier-melt contribution to river discharge can be quite important for various rivers in the region. Therefore, for this area the Global

Flood Awareness System version 4.0 (GloFAS 4.0) dataset is used as reference dataset (Harrigan et al., 2020; Grimaldi et al., 2022). GloFAS is a state-of-the-art hydrological model and observations from this region have been used for calibration and evaluation of the model (see Copernicus Emergency Management Service, 2025).

## 2.4    Experiment setup

### 2.4.1    Point-scale simulations

The point scale simulations over the PROMICE stations described in Sect. 2.3 are run for different periods for each site, depending on the time span of the available observations. To reduce spin-up issues, each site is run multiple times over the





available time period ensuring that at least 30 years of spinup are performed before running the final simulation used in the evaluation. Three types of experiments are run to evaluate the impact of the new glacier parameterisation depending on the forcing used to drive the model, as well as the glacier mask used to identify the glacier points.

For the baseline configuration, ecLand is run in stand-alone mode forced by a dataset that combines in situ meteorological observations with ERA5 reanalysis data (Hersbach et al., 2020). Specifically, the in situ observations of air temperature, humidity, wind speed and direction, surface pressure and incoming shortwave and longwave radiation are used to force the model; the solid and liquid precipitation data were sourced from ERA5. When missing values are present in the in situ observations, those are gap-filled using ERA5 data. This is a common approach for gap-filling of long time-series of

observational data (Morin et al., 2012; Essery et al., 2016). This may generate local inconsistencies but it allows to run a land-surface model with a consistent physical state in the snow-ice-soil column throughout the period. For all points, the glacier mask is set to $f_{\rm gl} = 1$, as the in situ observations are taken over the ice sheet. The baseline simulations will be referred to as "CTL-OBS" for the control experiment, using ecLand at CY49R1, and "GLA-OBS" for the experiment using the new parameterisation.

The second type of experiments uses all meteorological variables from ERA5, whilst the glacier mask is still based on the local in situ conditions ($f_{\rm gl} = 1$ for all points). These simulations will be referred to as "CTL-E5" and "GLA-E5" for the control and glacier experiments, respectively.

    The third type of experiments uses all meteorological variables from ERA5, and the glacier mask is based on the nearest neighbour grid point in the glacier mask used operationally at ECMWF, at 9 km resolution. These would be referred to as

"CTL-E5-CLIM" and "GLA-E5-CLIM", for the control and glacier experiments, respectively.

    The three types of experiments allow to evaluate the different impact of the new glacier parameterisation in an optimal setting (CTL-OBS and GLA-OBS), whereas the other two types allows to evaluate the impact of the new scheme in combination to forcing and glacier mask uncertainties, which is typically the condition when running the model globally at a coarser resolution using a numerical model output as forcing data.

**2.4.2   2D global simulations**

Global land-surface simulations have been run covering the time period from 1970 to 2022. In the global configuration, the model is forced with meteorological variables of the ERA5 reanalysis, the latter regridded in a conservative way to the target grid. The grid point resolution of the land-model is set to about 14 km (TCo799 on the original reduced Gaussian grid), whereas CaMa-Flood runs at a resolution of 15 arcmin, about 28 km at the equator, on a regular latitude-longitude grid. Although this

resolution is relatively coarse for hydrological applications, it represents a reasonable balance between computational cost and the efficiency of running long-term global simulations.



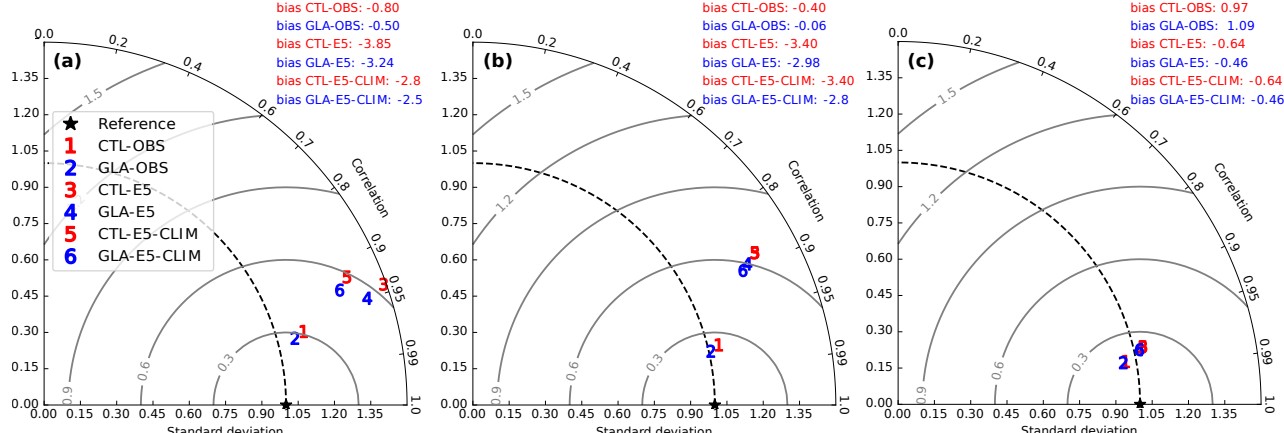

**Figure 2.** Taylor diagrams of hourly skin temperature for ecLand control experiment (CTL) and ecLand with the new glacier parameterisation (GLA) using in situ forcing (CTL-OBS and GLA-OBS), ERA5 forcing (CTL-E5 and GLA-E5) and ERA5 forcing and the ECMWF's operational glacier mask (CTL-E5-CLIM and GLA-E5-CLIM) compared to observations at the PROMICE stations, aggregated at the (a) *lower*, (b) *upper* and (c) *accumulation* sites. See Sect. 2.4.1 for details on the different experiment setups.

## 3 Results

### 3.1 Point-scale evaluation

#### 3.1.1 Surface temperature

An overall evaluation of the impact of the new glacier parameterisation (GLA), compared to the Control (CTL), on the hourly surface (skin) temperature simulated by all experiment types is reported in the Taylor diagrams of Fig. 2. Surface temperature RMSE and variability are generally improved with the new scheme, irrespective of the forcing or glacier mask used. The Taylor diagrams clearly indicate the large difference in model performances between simulations forced with in situ data (CTL-OBS, GLA-OBS) compared to those driven by ERA5 reanalysis (CTL-E5, GLA-E5) and the one using for $f_{gl}$ the value of the

nearest neighbour grid point of the ECMWF's operational glacier mask (CTL-E5-CLIM and GLA-E5-CLIM), highlighting the impact of forcing and ancillary data quality and the resolution of those in the model accuracy. ERA5 forced simulations at the PROMICE locations have larger RMSE than the ones forced with observations, due to a larger bias at the *lower* and *upper* sites. Interestingly, at the *lower* sites E5-CLIM simulations have a smaller bias than the E5 simulations, which is related to the seasonal variability of the bias at those sites, as will be discussed below. The smaller differences between CTL and GLA

experiments at the *accumulation* sites than at the other sites is due to these points being located at higher elevations on the Greenland ice sheet, where albedo and snow properties variations are less pronounced throughout the year and so differences among GLA and CTL are expected to be minimal.

The annual cycles of the monthly mean surface temperature bias at the PROMICE sites (see Fig. 3) show that both CTL-OBS and GLA-OBS exhibit a consistent cold bias of approximately 1 K throughout the year at both the *lower* and *upper*



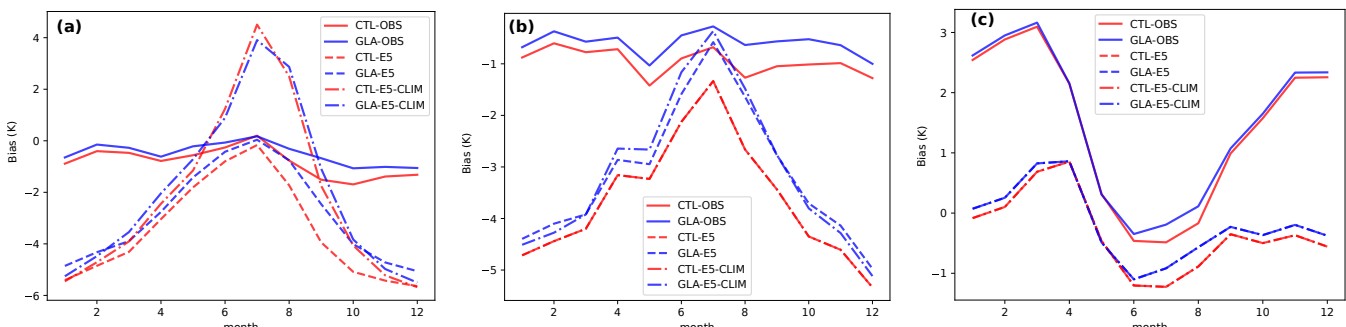

**Figure 3.** Mean annual climatology of modelled surface temperature bias relative to PROMICE observations at the (a) *lower*, (b) *upper* and (c) *accumulation* sites, for the ecLand control experiment with in situ forcing (CTL-OBS), with ERA5 forcing (CTL-ERA5) and with ERA5 forcing and ECMWF's operational glacier mask (CTL-ERA5-CLIM) and ecLand with the new glacier parameterisation (GLA-OBS, GLA-ERA5 and GLA-ERA5-CLIM for in situ, ERA5 forcing and ECMWF's operational glacier mask, respectively). See Sect. 2.4.1 for details on the different experiment setups.

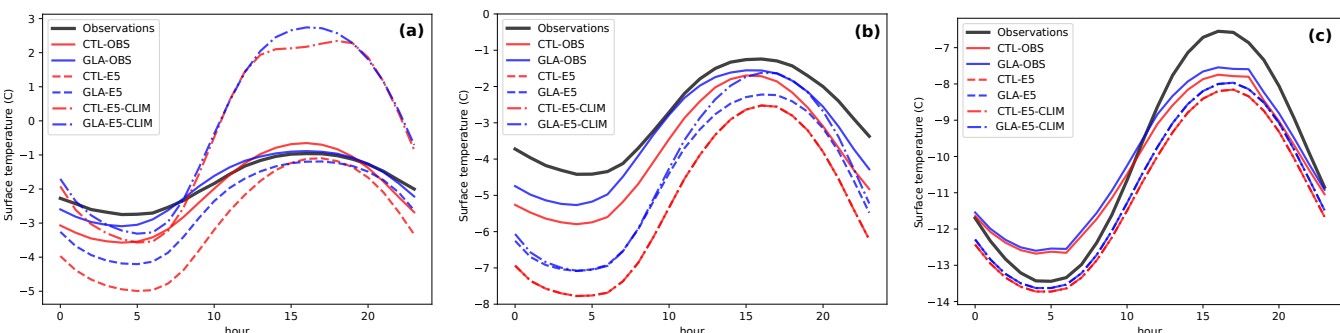

**Figure 4.** Diurnal cycle of surface temperature (°C) from PROMICE observations (black line) at (a) *lower*, (b) *upper* and (c) *accumulation* sites, compared to the one simulated by ecLand CTL with in situ forcing (CTL-OBS), and with ERA5 forcing (CTL-ERA5) and with ERA5 forcing and ECMWF's operational glacier mask (CTL-ERA5-CLIM) and ecLand with the new glacier parameterisation (GLA-OBS, GLA-ERA5 and GLA-ERA5-CLIM for in situ, ERA5 forcing and ECMWF's operational glacier mask, respectively). See Sect. 2.4.1 for details on the different experiment setups.

sites (Fig. 3a and Fig. 3b, respectively). Conversely, at these sites both CTL-E5 and GLA-E5 show a seasonality of the bias, with a bias of about -4 to -6 K in the winter months, which is significantly reduced during summertime. CTL-E5-CLIM and GLA-E5-CLIM, show further differences compared to CTL-OBS and GLA-OBS in the summer months at the *lower* sites, with a positive bias of about 4 K. The compensation between the cold bias in the winter and the warm bias in the summer months for the CTL-E5-CLIM and GLA-E5-CLIM experiments explains the overall lower bias compared to CTL-E5 and GLA-E5

reported in Fig. 2a.





The biases are generally reduced in all configurations in the GLA experiments, the largest impact being during the summer months. The *accumulation* sites show a different behaviour of the annual cycle of biases. At these sites the CTL-OBS and GLA-OBS experiments show the largest errors during the winter months, with a warm bias of about 3 K during wintertime, which might be associated with challenges of land-surface models to simulate stable boundary layers in the polar night (Dutra et al., 2015). On the contrary, CTL-E5 and GLA-E5 have biases of less than 1 K, highlighting the compensating effect of using the ERA5 data instead of observations to force the land-surface model.

At the *lower* and *upper* sites the differences in the surface temperature between the GLA and CTL experiments during the summer months are related to the dynamical albedo and the variations in the internal snow properties, which are most active during this period, hence impacting the diurnal cycle variability of the surface temperature. Figure 4 shows the diurnal cycle of surface temperature for the summer months, including all hourly data, indicating that the new glacier parameterisation improves the representation of the maximum, minimum and phase of the diurnal cycle at the PROMICE sites at the *lower* and *upper* sites. At the *lower* sites the improvement in the amplitude of the diurnal cycle is about 1 K for the observation- and ERA5-forced simulations, whereas this is less evident in the GLA-E5-CLIM experiment. This is due to those sites being located near the coast, and the low resolution glacier cover is hardly representative of the location of the station, with an average glacier coverage at the *low* sites of 0.60. This explains the large bias occurring in both CTL-E5-CLIM and GLA-E5-CLIM experiments during daytime, as incoming solar radiation is absorbed by the bare soil surface and the surface temperature is not limited by the presence of snow or ice. Conversely, at the sites further inland sites the improvement is consistent across the different forcing and glacier cover used, indicating that those sites, located more inland and at higher elevation, the low resolution glacier mask is more representative of the actual conditions.

The comparison of the different experiment types illustrates the importance of forcing and ancillary data for the model accuracy as well as for modelling development, which has previously been highlighted by several studies (Dutra et al., 2011; Réveillet et al., 2020; Terzago et al., 2020). In the context of modelling development, Fig. 3 and Fig. 4 indicate that biases are tightly linked to the type of forcing and ancillary data used. Therefore, the choices in parameter tuning or process developments should account for forcing and/or ancillary data uncertainties to target the specific problem that is under investigation.

### 3.1.2 Snow temperature

The improvements in the simulation of the surface temperature also propagates into the subsurface, affecting the deeper snow temperatures. For conciseness, only CTL-OBS and GLA-OBS are reported below, as the results for E5 and E5-CLIM experiments are consistent with the ones reported. Figure 5 shows the Taylor diagrams of hourly snow temperature at one metre into the modelled and observed snowpack for the PROMICE stations at the *upper* and *accumulation* sites; *low* sites are not shown as the snowpack can melt completely at those sites during the summer months, and therefore making a consistent comparison at the same depth between the model and observations difficult. GLA-OBS shows a large improvement compared to CTL-OBS, in particular at the *upper* sites, reducing a cold bias of several degrees in the CTL experiments. To better illustrate the increased realism in the snow internal processes, Fig. 6 shows the time series of daily averaged snow temperature at 1 m depth at the TAS-U site, together with liquid water content in the snowpack accumulated in the first metre at the same site.



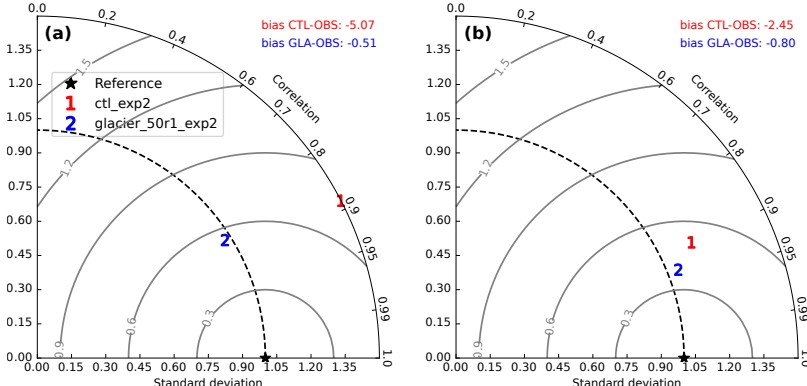

**Figure 5.** Taylor diagrams of hourly snow temperature at one metre into the modelled snowpack for ecLand control experiment (CTL) and ecLand with the new glacier parameterisation (GLA) using in-situ forcing (CTL-OBS and GLA-OBS) relative to observations at the PROMICE stations, aggregated at the (a) *upper* and (b) *accumulation* sites. See Sect. 2.4.1 for details on the different experiment setups.

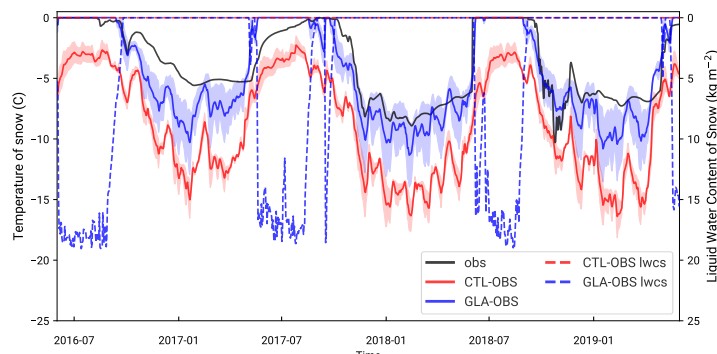

**Figure 6.** Time series of daily averaged snow temperature at one metre into the observed snowpack at the TAS-A site (black), the modelled snow temperature interpolated at one metre (continuous lines), and the accumulated snow liquid water content in the snowpack accumulated over 1 m (dashed lines) for the CTL-OBS (red) and GLA-OBS (blue) experiments. The shaded area indicates the snow temperature range between the 0.75 m and 1.25 m depths.

CTL-OBS never reaches the melting point during summertime, whereas GLA-OBS shows a more realistic range of variations of the snow temperature. The change in snow temperature is also reflected in the liquid water content, which increases during summer as a result of the ice melting in the snowpack. Part of the liquid water refreezes during nighttime or at the end of the summer period, releasing latent heat and warming the snowpack.





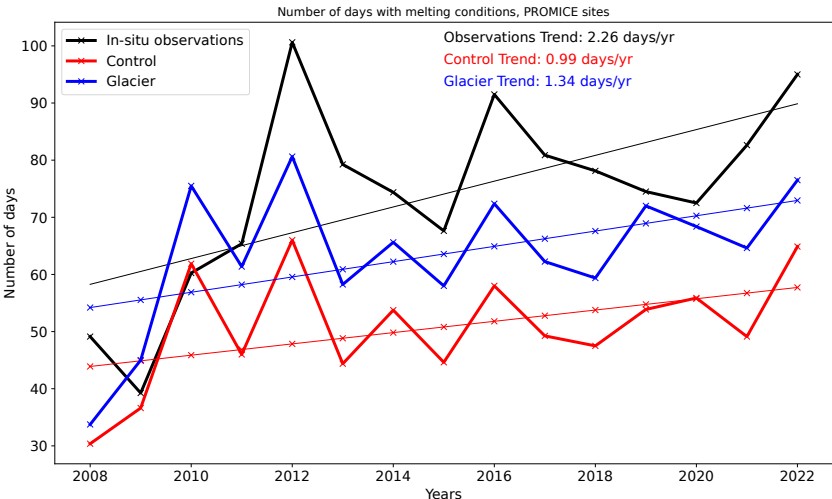

**Figure 7.** Time series of yearly snowmelt occurrences averaged over all the PROMICE stations (black), and modelled in the ecLand control (Control, red) and ecLand with the new glacier parameterisation (Glacier, blue) experiments. All sites with observations covering the period 2008 to 2022 are considered in the analysis.

### 3.1.3 Trends in melting occurrences

Previous sub-sections have focussed on the improvements in the representation of the surface and subsurface temperature as well as snowpack properties. Accurately modelling the snowpack and surface temperature is essential for predicting the timing of melt events. To quantify how snowpack and ice modelling improvements can affect melting event diagnostics, Fig. 7 presents the time series of yearly melt occurrences, averaged across the PROMICE stations with available data between 2008 and 2022. A melt event is identified when the surface temperature exceeds the melting point of water. The in situ observations show a

statistically significant increase in the number of melt occurrences over the period, with a trend of 2.26 days yr$^{-1}$ (p-value $< 0.05$). CTL-OBS largely underestimates the number of melt occurrences, both the frequency of melt events and the overall trend in melt occurrences during the analysed period, the latter not being statistically significant at the 95% confidence level. GLA-OBS generally demonstrates a better agreement with the observations and shows a slight improvement in capturing the trend of the melting occurrences. Even though underestimated, the trend in GLA-OBS is statistically significant at the 95%

confidence level, with a value of 1.34 days yr$^{-1}$ (p-value $< 0.05$). Part of these remaining differences might be due to a too coarse vertical discretization of the snowpack (0.50 m for the layer in contact with the atmosphere), which may lead to overestimation of the thermal inertia of the snowpack, not allowing the surface to respond (i.e. warm) quickly to changes in the atmospheric radiative forcing.



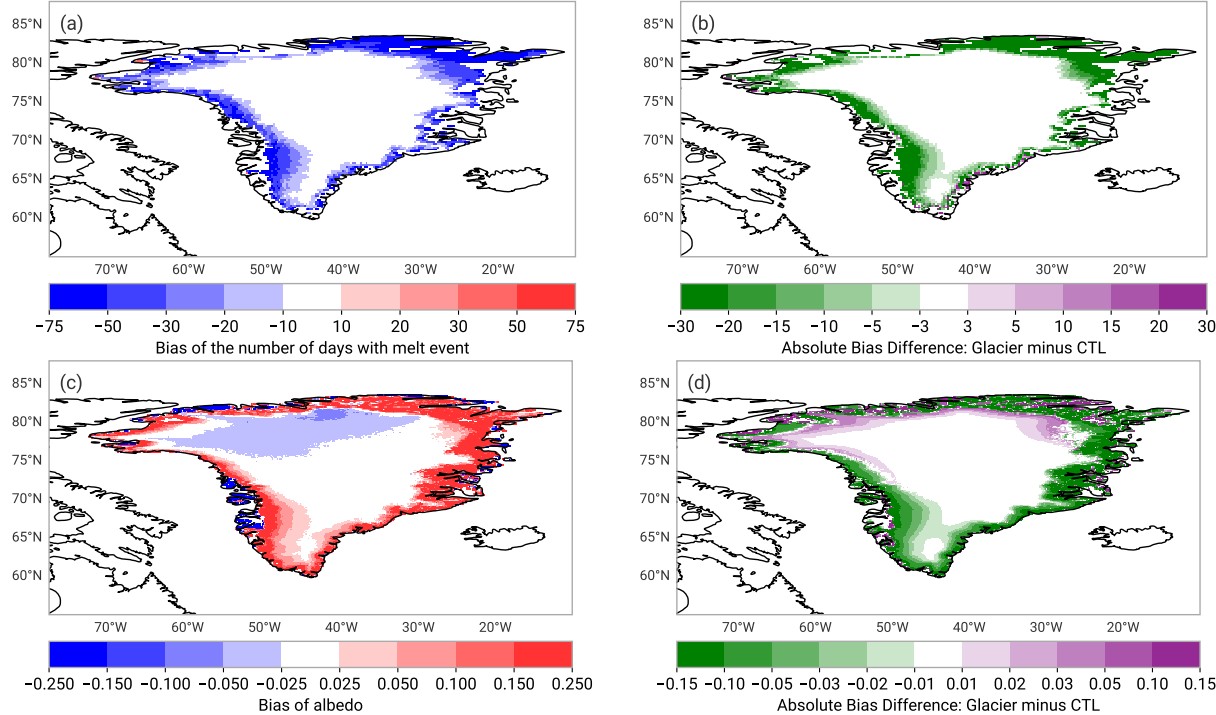

**Figure 8.** (a) Difference (bias) in the yearly average melt occurrences from 1990 to 2012 between MEASUREs product and the CTL experiment and (b) the difference of the absolute bias in melt occurrences between the GLA and CTL experiments; (c) Difference (bias) in the summertime (June to August) monthly climatology of albedo (from 2000 to 2012) between the MODIS product and the CTL experiment and (d) the difference in the absolute bias of albedo between the GLA and CTL experiments for the same period. In panels (b) and (d), green shadings indicate the regions where the magnitude of the bias is reduced in GLA compared to CTL.

## 3.2 Regional-Scale Evaluation Over Greenland

### 3.2.1 Melt Occurrences and Albedo

Section 3.1.3 highlighted the improvements in simulating melting occurrences at the PROMICE sites. To generalise these findings at the regional scale, a satellite-derived dataset of melt occurrences and a MODIS-based albedo product are used. Melt day occurrences in Greenland are evaluated using the MEaSUREs dataset, which provides daily melt/no-melt classifications of the ice sheet (see Sect. 2.3). The dataset is derived from changes in the passive microwave brightness temperature of the snow in the presence of liquid water in the snowpack, and therefore the presence of liquid water in the top snow layer would be the best indicator to compare with this product. However, as CTL does not compute liquid water presence over glaciers, melt occurrences in the model are diagnosed using the top layer snow temperature exceeding the melting point.

Figure 8a shows the difference in the yearly average melt occurrences from 1990 to 2012 between the MEaSUREs product and the CTL experiment, indicating that CTL largely underestimates those around the coast of Greenland. This is greatly





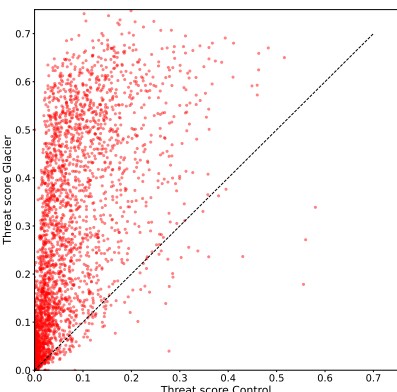

**Figure 9.** Daily threat score of melt occurrences over the Greenland ice sheet from 1990 and 2012 for the CTL (x-axis) and GLA (y-axis) experiments. The threat score is calculated relative to the MEaSUREs Greenland Surface Melt Daily dataset following de Rosnay et al. (2015).

improved in GLA (see Fig. 8b), which shows a general increase (that is a reduction of the bias compared to CTL) in the melt occurrences during the year. The main contribution to this behaviour is the albedo, which over the coastal region is overestimated during summertime (June to August average) in CTL compared to the MODIS product, as shown in Fig. 8c. The bias in albedo is reduced in GLA (as indicated by the reduction in the difference of the absolute value of the biases in Fig. 8d), indicating lower albedo values with the new glacier scheme, which allows for more energy to be absorbed by the surface and

therefore more melting occurrences. It is worth noting that the CTL shows a negative bias in the northern part of Greenland (Fig. 8c), implying an underestimation of albedo in this area. This is slightly degraded in GLA in the transition zone between the coast and the accumulation zone (Fig. 8d). This effect may be linked to a feedback mechanism between snow albedo and temperature in the new albedo formulation, which can be triggered in dry regions with limited solid precipitation, such as northern Greenland (see for instance Box et al., 2004). According to Eq. 3, an increase in snow temperature reduces albedo,

causing more shortwave radiation to be absorbed at the surface, which in turn increases the temperature and so further reducing the albedo. Fresh snowfall would restore the albedo to higher values, but in areas with low snowfall the resetting process is absent allowing the feedback to persist and sustain lower albedo values over time. A formulation of snow albedo based on the snow microphysical properties (see for instance Zorzetto et al., 2024) might be more appropriate to avoid such feedback mechanism, as the snow albedo would be directly related to the snow grain size or optical depth rather than to the temperature

of the snowpack.

    A more detailed quantitative comparison of melt occurrences is presented in Fig. 9, which illustrates the daily threat score of melt occurrences across the Greenland Ice Sheet from 1990 to 2012. The threat score is calculated, following de Rosnay et al. (2015), at each grid point relative to the MEaSUREs product, providing an integrated measure of the model's ability to accurately simulate melt events. GLA demonstrates a significant improvement in the threat score, as evidenced by the majority





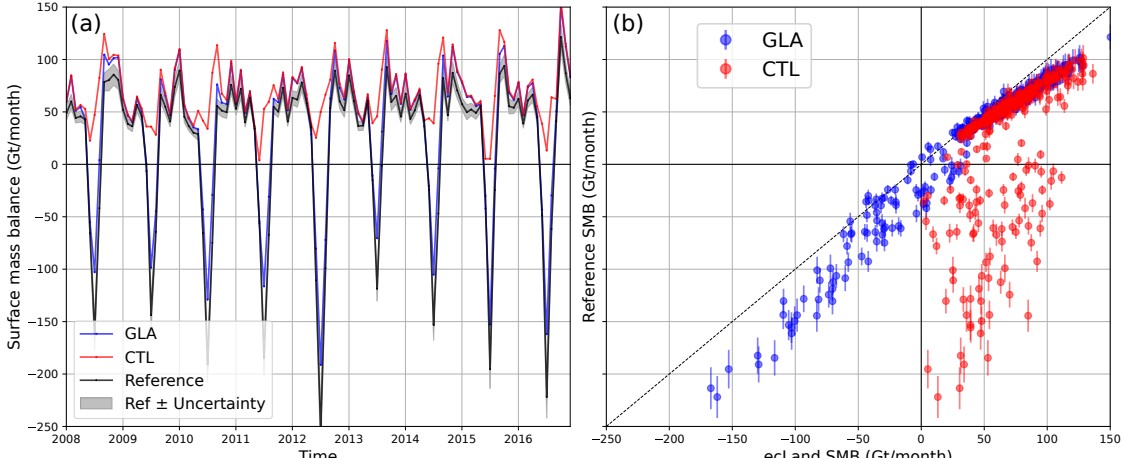

**Figure 10.** (a) Time series of surface mass balance (Gt/month) for the Greenland ice sheet between the reference dataset (Otosaka et al., 2022, black line) and the CTL (red line) and GLA (blue line) experiments, for the period 2008-2016; the shaded grey areas indicate the uncertainties in the reference dataset. (b) Scatter plot comparing the surface mass balance (Gt/month) between the reference dataset, on the y-axis, and the CTL (red) and GLA (blue) experiments, on the x-axis, for the period 1990 to 2020; vertical lines indicate the uncertainties in the reference dataset for each month.

of data points lying above the 1-to-1 line on the scatter plot. This indicates that the experiment using the new scheme more accurately predicts melt occurrences compared to the control simulation across the Greenland ice sheet, particularly in regions where melting is prevalent.

### 3.2.2   Surface Mass Balance

The previous sub-sections demonstrated the improvements in simulating the surface temperature and melting occurrences over
the Greenland ice sheet. These improvements are expected to significantly influence the surface mass balance (SMB) of the ice sheet; meltwater can percolate into the snowpack and either refreeze, contributing to internal ice processes, or exit the snowpack as runoff, directly impacting the ice sheet's mass balance.

    Figure 10 shows the comparison of the surface mass balance (Gt/month) for the Greenland ice sheet as computed in the reference dataset (see Sect. 2.3) and the CTL and GLA experiments. The time series in Fig. 10a displays only a subset of
the available time period, from 2008 to 2016, to visually compare the different model outputs, whilst a more quantitative comparison is shown in in Fig. 10b using data for a longer time period (from 1990 to 2020).

    The annual cycle of the surface mass balance indicates a long accumulation period from autumn to spring, followed by a melting period mainly confined to the months July/August (see Fig. 10a). CTL shows a large underestimation (in absolute terms) of the ablation period, with SMB always positive throughout the year (see Fig. 10b). For SMB values exceeding
60 Gt/month, CTL exhibits a positive bias likely due to an overestimation of accumulation during wintertime (see Fig. 10a),



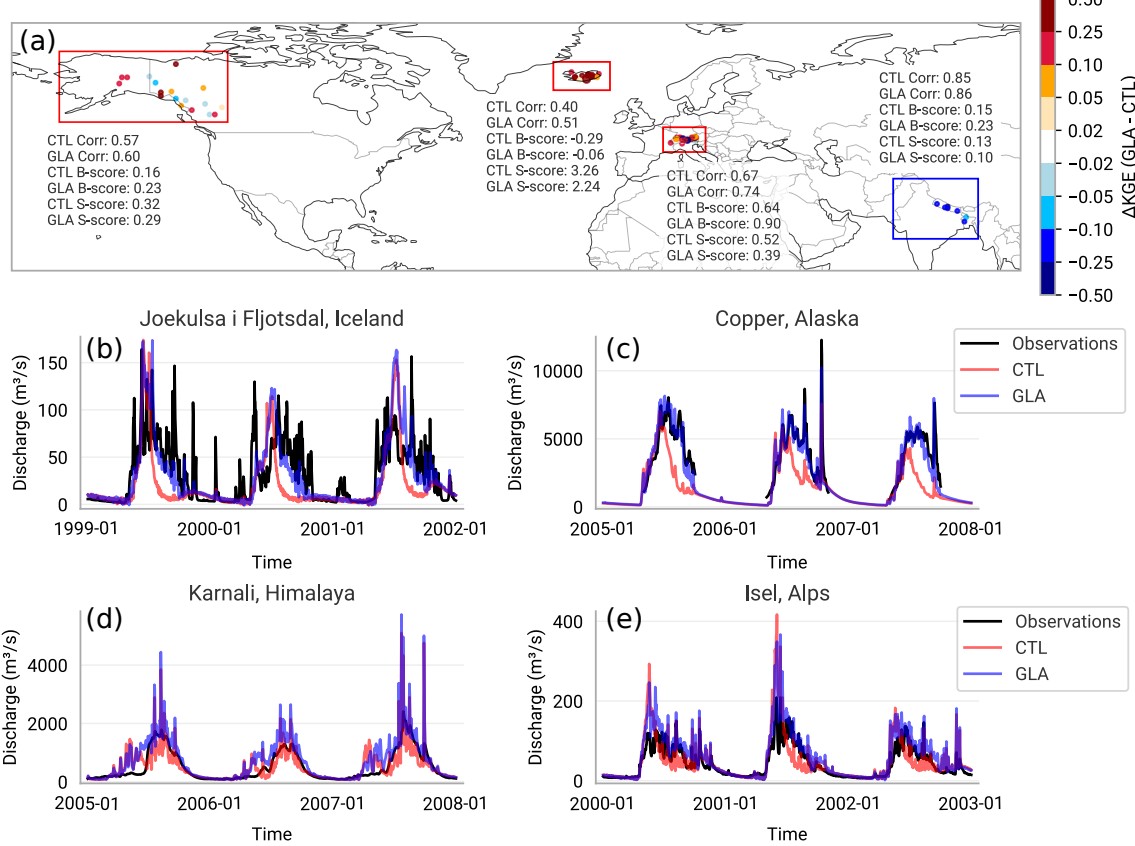

**Figure 11.** (a) Map of the Kling-Gupta Efficiency (KGE) for river discharge in different regions, comparing the new glacier parameterisation (GLA) with the control experiment (CTL). The time series of river discharge for selected gauge stations are shown in (b) for the Icelandic, (c) the North American, (d) Himalaya and (e) Alpine regions.

which can be due to ERA5 having higher snowfall rates over Greenland than the reference dataset. On the other hand, GLA shows a temporal evolution more consistent with the reference data, being able to capture the magnitude of melting during the summer months as well as the interannual variability. The GLA experiment still shows an underestimation (in absolute terms) of SMB during the ablation period, which can be due to forcing differences as well as to processes missing within the model, like wind-driven snow sublimation and firn-to-ice conversion.

## 3.3 Impact on River Streamflow

The evaluation of the new scheme for glaciers located outside the main ice sheets is a challenging task due to the representiveness of in situ observations and the coarse grid-box resolution of global simulations, meaning that most of the glaciers are not explicitly resolved in the model but covering a sub-grid fraction. To partly overcome these limitations and in particular the challenges of a direct comparison with in situ observations, the impact of the new glacier scheme on river streamflows which



are fed by glacier melt is evaluated. This allows to evaluate the impact of the new glacier scheme on the integrated hydrological cycle at the basin-scale, without the need of a direct comparison with local snow and ice observations.

Figure 11a shows the Kling-Gupta Efficiency (KGE, Gupta et al., 2009) for river discharge in different regions with significant glacier coverage (fully resolved or sub-grid), comparing the experiment using the new glacier parameterisation with the control experiment. The map includes summary statistics for all observations in the considered regions for the three components of the KGE (correlation, bias and variability) for the CTL and GLA experiments. Bias and variability are computed as bias score (B-score) and variability score (S-score) respectively, following Zsoter et al. (2022); a value of 0 for these scores indicates the optimal outcome, i.e. no bias and a perfect match in variability, respectively.

Results indicate an improvement in the KGE of GLA for the North American, Icelandic and European alpine rivers, and a degradation for the Himalayan rivers used in this study. Overall, the inclusion of the new glacier scheme increases the magnitude of the streamflow in the rivers during summertime, as a result of increased snowmelt and ice melt. This has a positive impact in terms of temporal dynamics and magnitude of the river discharge for river basins for which glacier melt is a significant contributor to the total streamflow during spring and summertime. This is illustrated by the time series at selected river gauges in the Alpine, North American and Icelandic regions (Fig. 11b,c and e), showing that CTL underestimates river discharge during summer months, as well as the interannual variability; these characteristics are better represented in the GLA experiment.

The comparison with the GloFAS dataset (see Sect. 2.3) for the Yarlung Tsangpo-Brahmaputra and Ganges basins in the Himalayan region shows a decrease in the KGE in GLA. At these gauges, the correlation between the modelled and observed discharge is increased in the GLA experiment, indicating improved dynamics of the river discharge. However, the increase in the positive bias (a degradation of B-score), in combination with a similar variability (S-score), contribute the most to the KGE score and so leading to a lower KGE in the GLA experiment. From the time-series of discharge for the Karnali river (Fig. 11d), both experiments overestimate the magnitude of the river flow during the spring months; however, this is more pronounced for GLA, suggesting an overestimation of snow and ice melt water with the new glacier scheme. The increase of the bias in the GLA experiment may stem from multiple factors. In terms of atmospheric forcing, ERA5 overestimates the solid precipitation in the Himalayan region (Orsolini et al., 2019), which could lead to an overestimation of snowmelt flux during the ablation period. In CTL this can compensate for the lack of the ice melting processes, whereas in GLA the combination of excess snow melt and ice melt can lead to an overall overestimation of the total meltwater. Moreover, the complex geography of the Himalayan region (e.g. the Karnali watershed's elevation ranges from 140 m to 7498 m) may be challenging to represent in a global simulation using a coarse resolution. Also the lack of representation of human activities in ecLand, like abstraction of water for irrigation or dam regulation for the Yarlung Tsangpo-Brahmaputra and Ganges basins, can contribute to the overestimation of the river discharge in the model. In particular water abstraction for irrigation purposes is particularly significant in these regions, reducing the observed total flow and potentially contributing to the bias in the modelled discharge. However, a quantitative attribution of these causes of errors is beyond the scope of this study and will be addressed in future work.





## 4 Conclusions

A new glacier parameterisation has been implemented in ecLand, which is the land-surface component of the Integrated Forecasting System (IFS), as part of the upcoming cycle 50R1. The new scheme consists of a land-ice tile and ice column over land, and further modifications to the snow scheme to allow for a more realistic representation of the snowpack properties over glaciers and ice-sheets. The new scheme has been evaluated offline (land-surface only simulation forced by in-situ or reanalysis data) at the point-scale at a range of sites of the PROMICE network over Greenland; at the regional scale, a range of different observational datasets and products targeting different processes have been used for a comprehensive evaluation of the model performance.

The new glacier scheme demonstrates improved accuracy in representing the surface and subsurface temperature at the PROMICE sites, due to an improved physical representation of snow and ice processes relevant over glacier surfaces. In particular, a dynamic albedo formulation accounting for decreased albedo for near-melting conditions, significantly improves the diurnal cycle of surface temperature in the summer months and the snow temperature of the deeper snowpack, at the sites located in the ablation zone of the ice sheet. These enhancements lead to a more accurate simulation of melting events over the Greenland ice sheet, which, combined with a consistent water budget calculation, allows for a more realistic estimation of the surface mass balance (SMB). The SMB computed by ecLand using the new scheme shows a good agreement with SMB estimates from state-of-the-art dedicated regional climate models (RCMs), highlighting the potential of using future reanalysis products for surface mass balance applications.

The results of the point-scale simulations indicate that forcing and ancillary data quality are crucial for model accuracy. Land surface model developments are usually designed and evaluated using in situ meteorological and surface process observations, with the aim of improving the model's description of the underlying physical processes. However, results of this study indicate that land-surface model developments that are then applied at regional or global scales, using reanalysis forcing data and coarse-resolution ancillary data, should account for forcing and ancillary data uncertainties, as model biases can be significantly affected by such uncertainties, not just in magnitude but even the direction (e.g. the sign of the bias). A balance between a model accurately describing local in situ conditions in an ideal setting and a model accounting for compensating errors from external sources should be pursued in the model development process, as the latter is more representative of the conditions when running the model in real-world applications.

The impact on river discharge from the new glacier scheme has been evaluated across four regions of the Northern Hemisphere with significant glacier coverage. The additional melting of snow and ice over glacier points leads to an increase of river discharge during late spring and summer months in basins where these processes are significant contributors to the total streamflow. The KGE of the river discharge is generally improved, indicating that the inclusion of the additional glacier-related processes has a positive impact on the integrated hydrological cycle at the basin-scale. This comes mainly from an improved correlation with the observed discharge and a better representation of its temporal variability. However the magnitude of the simulated discharge may be degraded in regions where this is already overestimated in the control experiment; in the present study, this occurs for the stream gauges in the Himalayan region used in this study. This can be due to several reasons, among





others the lack of representation of human activities and dam regulations in the model, biases in the atmospheric forcing dataset,
as well as the use of a very coarse resolution (14 km and 28 km for the land-surface and river routing schemes, respectively)
which might introduce biases in the amount of snowmelt and ice melt ending in the river. Future work is required to further
investigate the sensitivity to the model resolution, in particular for applications in high-resolution future land reanalyses.

The explicit inclusion of land-ice surfaces opens new opportunities for advancing the representation of cryosphere processes
in ecLand. This includes the potential integration of a realistic firn column, firn-to-ice conversion, and an albedo parameterisation
based on snow microphysical properties. Moreover, while increasing the physical complexity of a land-surface model often
adds challenges to model calibration and tuning, incorporating previously missing physical processes, such as land-ice in this
case, may simplify the calibration. This is because in pre-existing model configurations, errors arising from missing processes
were often compensated for by tuning parameters that are not directly linked to the underlying physical processes responsible
for the error. Such compensation of errors can then affect other regions or climate conditions. Linking the errors to new
parameters associated with the added processes can make the model calibration more targeted, reducing unintended impacts in
other regions or climate conditions.

*Code availability.* The ecLand source code will be available in the ecLand GitHub repository
(https://github.com/ecmwf-ifs/ecland) as soon as IFS CY50R1 is operational.

*Data availability.* Data used in this work will be available upon publication on zenodo.org:
https://doi.org/10.5281/zenodo.15499505.

*Author contribution.* GA developed the code, designed the experiments and wrote the draft of the manuscript; CR contributed
to the writing; GB contributed to the initial design and the writing.

*Competing interests.* Authors declare no competing interests are present.





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
