# Peer review of "An improved glacier parameterisation for the ecLand land-surface model: local, regional and global impact"

_EGUsphere, 2025_

## Referee Comment (RC1)

An improved glacier parameterisation for the ecLand land-surface model: local, regional and global impact

The authors updated the parameterizations for glacier ice and snowpack within the land surface model and carried out validation over the Greenland Ice Sheet. The updated model demonstrated comparatively strong performance on the Greenland Ice Sheet and yielded valuable outcomes for the future development of ecLand. However, there remain several unclear aspects and areas lacking sufficient validation. I hope that the following comments will help improve the manuscript.

**Major comments:**

1. The differences between the existing model (CTL) and the updated model (GLA) are not clearly explained. For example, it is not clear what parameterizations the CTL used. Since this is a key part of the manuscript, I suggest showing the differences between the two models in a table to enhance clarity.

2. The transition from validating against the Greenland Ice Sheet to evaluating river discharge in the Northern Hemisphere feels somewhat sudden. Validation using in-situ albedo may be challenging, but would it be possible to compare the land surface albedo in CTL or GLA with MODIS albedo across the Northern Hemisphere? At least, the authors should discuss whether the GLA experiment also shows a decrease in albedo outside of Greenland.

**Specific comments:**

Title: In my impression after reading this manuscript, the use of the term "global" feels somewhat excessive. Since the river discharge validation is conducted only for the Northern Hemisphere, it might be better to change the title to "local and regional impact," or remove the phrase ": local, regional and global impact."

L35 (Surface processes…): Add melting and refreezing processes.

L67: What do you mean by "physical capping"? Please add a clearer explanation.

2 Methodology: It is better to briefly describe the difference between the GLA parameterization and the other LSM parameterization. Lee et al. (2024) might help you to compare GLA with LSMs (land surface models).

L87: Does "fully coupled" mean "Atmosphere, land and Ocean coupling"?

2.2 New glacier parameterisation: Although the GLA parameterizations are described, it is not clear what has changed compared to the CTL parameterizations. I would like to understand the differences between the two models before seeing results, so I suggest summarizing the characteristics of both models in a table.

L157-158: Previous studies have suggested that new snow density in the polar region exceeds 300 kg m-3 (Greuell and Konzelmann, 1994; Lenaerts et al., 2012; Niwano et al., 2018), so I have no objection to your assumption for the Greenland simulation. However, you should be careful if you apply the assumption to midlatitude areas. New snow density in midlatitudes is typically around 100 kg m-3 as indicated by previous studies (e.g., Niwano et al., 2012). Regarding this point, it should be stated whether changes in snow density parameterization affect snow albedo.

L169-170: Please add the references for the values you used for Eq. (3). If the values are not based on previous studies, please add the rationale why you set the values.

L192: If there is no alternative to the spatial SMB data other than RCM, it should be mentioned.

L205 (different periods for each site): This is vague explanation. Please describe clearly. It would be helpful if you could make a table summarizing the experimental setting, including other experimental settings. A supplemental material might be good.

L220: Did you apply elevation correction to ERA5?

2.4.2 2F global simulations: Please add an explanation regarding spinup simulations.

Figure2 (skin temperature): "Skin temperature" and "surface temperature" are mixed up in the main text. If they mean the same thing, please unify either.

Figure2: What period does the analysis in this figure cover? Please add the period and season for the analysis.

Results: The Results section should be nominally limited to new results from the current observation or calculation and not include a literature review (L264, 281. 334…). I found that the authors' interpretations are included within this section (e.g. L280-284, L332-340). I suggest that you change "Results" section to "Results and Discussion" section.

Line 242: Could you tell me about the specific scheme you improved?

Figure 3 (c): In the OBS experiment, temperature from PROMICE is used as an atmospheric forcing, yet panel (c) shows a bias of nearly 3°C during winter. The result looks strange. Please verify that there are no errors in the simulations or analyses. If no errors are found, the bias may be due to ERA5 precipitation used in the OBS experiment. Additionally, I could not locate the CLIM experiment lines in the figure.

Figure 4 caption: Add the specific season you analyzed.

Line 270: Add the specific months.

Line 280: This paragraph is clearly a discussion.

Figure 5: Please modify the legend. It looks like an old version. It is better to make the texts about the bias values larger.

Figure 6: I could not find the red dashed line at first. Please add that the red dashed line can be seen at 0 kg m-2.

Line 310: It is better to add the explanation regarding snow layers in the models to the method section.

Figure 8: As the color bar in the upper panels and the map in the bottom panels are close to each other, the labels on the color bar were misunderstood for the titles of the lower panels. In addition, please describe the difference between CTL and the validation dataset you analyzed to the labels of Figure 8a and c (for example, difference between Glacier minus CTL, like Figure 8b and d).

Figure 11(a): The text for CTL and GLA is cluttered, so why not red and blue text for CTL and GLA, respectively?

Figure 11(b-e): It is hard to see each line. How about changing the line style to a dashed line for CTL and GLA?

Line 393: Could the cause of this overestimation be the decrease in albedo in the GLA experiment? In the validation over the Greenland Ice Sheet, the GLA experiment showed a significant reduction in

albedo. It is necessary to compare the land surface albedo from MODIS, CTL, and GLA across the Northern Hemisphere and discuss whether the new parameterization leads to a decrease in albedo over the Northern Hemisphere.

References:

Greuell, W. and Konzelmann, T.: Numerical modelling of the energy balance and the englacial temperature of the Greenland Ice Sheet. Calculations for the ETH-Camp location (West Greenland, 1155ma.s.l.), Global Planet. Change, 9, 91–114, https://doi.org/10.1016/0921-8181(94)90010-8, 1994.

Lee, W.Y., Gim, HJ. and Park, S.K.: Parameterizations of Snow Cover, Snow Albedo and Snow Density in Land Surface Models: A Comparative Review. *Asia-Pac J Atmos Sci* **60**, 185–210, https://doi.org/10.1007/s13143-023-00344-2, 2024.

Lenaerts, J. T. M., van den Broeke, M. R., Déry, S. J., van Meijgaard, E., van de Berg, W. J., Palm, S. P., and Sanz Rodrigo, J.: Regional climate modeling of drifting snow in Antarctica, Part I: Methods and model evaluation, J. Geophys. Res., 117, D05108, https://doi.org/10.1029/2011JD016145, 2012

Niwano, M., Aoki, T., Kuchiki, K., Hosaka, M., and Kodama, Y.: Snow Metamorphism and Albedo Process (SMAP) model for climate studies: Model validation using meteorological and snow impurity data measured at Sapporo, Japan, J. Geophys. Res., 117, F03008, https://doi.org/10.1029/2011JF002239, 2012.

Niwano, M., Aoki, T., Hashimoto, A., Matoba, S., Yamaguchi, S., Tanikawa, T., Fujita, K., Tsushima, A., Iizuka, Y., Shimada, R., and Hori, M.: NHM–SMAP: spatially and temporally high-resolution nonhydrostatic atmospheric model coupled with detailed snow process model for Greenland Ice Sheet, The Cryosphere, 12, 635–655, https://doi.org/10.5194/tc-12-635-2018, 2018.

---

## Referee Comment (RC2)

Review of " An improved glacier parameterisation for the ecLand land-surface model: local, regional and global impact"
by
Gabriele Arduini, Christoph Rüdiger, and Gianpaolo Balsamo

The authors present results of evaluation tests of a new tile within the ecLAND module of the IFS-code of the ECMWF. I have read this paper with interest - the paper is generally well written - but needs improvement in the depth of the evaluation. The authors now convincingly show that the code update is an improvement. However, it would be good if the authors show also the performance against existing models, to see if the new tile/module performs similar as those models, or that there is still room for further improvement.

The essential improvements are those in the evaluation of the skin temperature, and especially of those of the subsurface temperature and to add an evaluation of in situ surface mass balance. I would genuinely like to see the technical improvements that I think are essential for better representation of the SEB and SMB without much coding - at the other hand I am afraid this request/wish will be in vain.

Comments
Introduction: glaciers and ice sheets are two different things, so by using "glaciers", ice sheets are sometimes forgotten. Therefore, I think it is better to replace
"glaciers" of lines 24, 32, 39, 106 to "glaciers and ice sheets",
"glaciers" of lines 30, 94, 97, 100, 107 to "glaciated surfaces".
L 108 is a special case, I leave it to the authors to make it more general. And check other instances when I have missed the word "glacier" in this list.

L 34: Greenland Ice Sheet - so with capitals.

L 38: Don't forget that for ice sheet, Earth System Models (like CESM2, UKESM) perform increasingly well (if the ESM developers try to model the SMB well), so regional climate models are not the only alternative of using observations only. These things are discussed later in the introduction too, so this formulation is already somewhat inconsequent.

L 43-54: I don't think this sidestep to ice dynamics is relevant for this manuscript nor introduction. For example, the initial-condition problem of ice sheet models is completely different to those of atmospheric models. I propose to remove this paragraph.
L 48: I think this reference if outdated, take a newer one if you decide to retain this paragraph.

L 65: A regional climate model is not an Earth System model. So, HARMONIE-AROME improvement are not an example of ESM improvements.

L 74-82: Given that the land-ice parameterization of ecLand is simple and uses very few layers compared to the advanced schemes in, particularly, polar adapted Regional

Climate Models, I would like that the authors formulate very specifically what their new parameterization should do well, and what not necessarily. For example, is that the SEB (including albedo evolution) for the typical range of a weather forecast (14-21 days), on seasonal timescales; does it include the surface mass balance (including refreezing effects)?

L 94: As far as I can recall, this simple snow scheme has been used also in earlier IFS cycle. If so, replace by "In CY49R1 (the NWP ....2024) and preceding versions since version <first version with this code>, glaciers and ice sheets are ...."

L 135: This method of confining the skin temperature to the melting point is already long in the IFS code even before Arduini 2019. Still, I remain to the opinion that this workaround is a poor solution, given that one can easily solve this issue in a mathematically sound and numerical simple way. We have used previous IFS versions (e.g. CY33R1) and in those versions considerable errors arose. Specifically, the time step after observing melt, the initial skin temperature guess was below the freezing point, the large conductivity number was not used, and the actual skin temperature became again well above the melting point. Even if this problem does not arise in this version, the method of Arduini leads to inconsistent skin temperatures if the uppermost snow or ice layer is not at the melting point.
Our solution was and is the following: When one observes that the skin temperature of a snow/ice/glacier tile is above the melting point, we keep the skin conductivity as is, but (simply) apply that one thus knows that the skin temperatures of that tile is at the melting point. If those tiles cover the whole grid box, one only need recalculate the fluxes. If these melting surface cover only a part of the grid box, we solve the SEB again, applying that we know the skin temperature of these melting snow/ice tiles, while the other tiles remain unknown as in the normal linearized SEB solve method. The authors can have more extensive documentation of our approach if they wish to have.

Section 2.2.2:
To me, this decision to not create separate prognostic variables for glaciated variables is a very, very poor compromise. For the glacial ice it is acceptable - sea ice is not active over land - but for snow I really do not see the urgency to mess up your snow physics to safe 4x4 prognostic variables (layer mass/thickness, layer density, layer temperature, layer water content). Snow surfaces, and particularly the snow albedo, is a classic example of a non-linear process, so the aggregate of two implies that now both are wrong.
We have no longer the computers of the 70s for which fast memory was a severe limiting factor and is the fancy type system of IFS not specifically set up to allow for adding new variables without having to adjust the code from top-to-bottom?
It sounds like that the authors were allowed to play around and improve the representation of glaciated surfaces; under the condition they won't bother the rest of the ECMWF-IFS community in any way. Don't get me wrong, I applaud the efforts of the authors to improve the representation of glaciated surfaces, but this compromise is very typical of the general and decades-long neglect of glaciers and ice sheet surfaces by the ECMWF.

Ideally, this poor compromise is rectified, and glacial snow is separated from land snow (and if you are doing that, please also separate snow over low vegetation and the snow below high vegetation into two independent snow layers sets, as that mix-up is equally bad). But I also do understand this strong suggestion (separate variables) is infeasible to effectuate (now), and I am aware this rant leads to zero change on the code or paper. However, hopefully it encourages the authors (or their successors) to fight even harder for a proper representation of glaciated surfaces in the IFC code. In all cases, I would like to see a longer motivation in the rebuttal why this poor compromise has been taken.

L 168: Please specify what Tsn is. It could be the temperature of the uppermost snow layer or the glaciated surface skin temperature - I don't know now.
I presume the authors are aware that this snow albedo (Eq. 3) includes two major simplifications. Firstly, the snow albedo is not a function of temperature (or density-per sé), but of snow grain size (and to some extend to the solar zenith angle). Secondly, the snow albedo is very strongly dependent on the wavelength, almost always 1 for UV to yellow; strongly varying for "red" and zero-ish for near-infra red - see e.g. Gardner and Sharp, GRL, 2010, or Van Dalum et al, The Cryosphere, 2020). So, using one albedo value number is similarly bad as running ecRAD with only one G-band. Again, I know it is not realistic to expect the authors to use a state-of-the-art grain-size based snow albedo scheme, but it is good to mention in the manuscript that this albedo is not regarded as state-of-the-art and specify its limitations with a reference to Gardner and Sharp, GRL, 2010 or a similar paper.

Section 2.4: It doesn't become completely clear to me how the experiments are carried out. This holds for both the point-scale as 2D global simulations. I understood from the description that the land model is rerun, but the atmospheric model not. Whether I got it right or not, explain in more detail which parts of the code have been rerun and which not. Furthermore, please specify which fields/fluxes are updated/adjusted in the experiments, and which fields/fluxes were kept constant. I conclude from the paper that the SEB has been recomputed, but I don't understand how that is done as that is far from trivial to do afterwards offline within the IFS framework (e.g. one needs the derivatives of fluxes to close the SEB and I would be seriously surprised as these derivatives are available from the ERA5 simulation.).

Figure 1 & 8: please use a map projection that shows Greenland with the right width to length ratio, so either Lambertian, Polar Stereographic or rotated lat-lon.

L 175: From the PROMICE dataset, the authors use the skin temperature. However, this temperature is not measured. I presume that the authors use the temperature derived from the upwelling longwave radiation. Please specify that explicitly here.

L 220: Make this description of the CLIM data more specific. Fgl is thus not 1 - otherwise that would have been stated - but still it would be sensible to use the tiled skin temperature for Figure 2-4. State which values have been used - grid-box average skin temperature or tile skin temperature. From line 275 I conclude that grid box averaged skin temperatures are used, it is better to adjust this - as the paper is about evaluating the glacier tile and the observations are on the ice sheet. So, why is not the tiled skin

temperature analyzed? This parameter can be exported, so practical reasons are not impeding this.

I like the analysis in Figures 2-4 - it is a good method to show what the new module can and cannot. The drawback is that it is hard to compare how well this new module performs compared to existing glacier surface descriptions, as I haven't seen it in other papers. Therefore, assuming that the PROMICE skin temperature is derived from the upwelling longwave radiation, the golden standard is the modelled skin temperature with a SEB model (e.g. https://doi.org/10.1017/jog.2024.68 ). It would be good add the performance of such a dataset, to compare it against the CTL-OBS and GLA-OBS results. Similarly, the "E5" and "E5-CLIM" could be compared of the perfomance of a polar adapted RCM like MAR (https://doi.org/10.5194/tc-14-957-2020) or RACMO (https://doi.org/10.5194/tc-12-811-2018 - although this dataset is outdated). I am quite sure the required data for such an analysis is available for the authors to be used.

L 252-260: The interpretation of Figure 3 is complicated without analysing the SEB and the T2m. An T2m temperature analysis could indicate if the atmospheric conditions of the E5 and E5-CLIM simulations are colder/warmer than those in the observations. Similarly the SEB analysis (against the PROMICE data) could indicate why the conditions are colder than observed. These figures don't need to be added to the manuscript, but such an analysis add depth to paragraph, which is now not much more than "Hmmm, our model is warmer/colder than observed." In that respect, the 3K warm winter bias of both OBS is remarkable, as for this experiment the 2m is "correct". In short, figure out why the new and old model deviate, and report that in the revised manuscript.

Figure 4: add in the caption that summer months are evaluated.

Concerning Figure 3-4, to which extend are the difference due to elevation difference between the observational site and the height of the grid box? Elevation biases induces temperature biases which have nothing to do with poor functioning parameterizations. Remove the effect of an elevation bias (if present).

Conserning Figure 4, again I would like to challenge authors to dive a bit more deeper into the 'why' the model is deviating from the observations. Very little physical explanations are given, and the manuscript (and your understanding of the model performance) will be improved if this indepth analysis is made. Again, the avenue to get this insight is through analysing the SEB, and again these figures don't have to be added to the manuscript (possibly supplementary materials), but allows for a more physical explanation why the model is deviating. I would guess that the underestimated cycle for the high locations is due to too high effective thermal capacity of the snow layer (being 50 cm thick), while alternatively missed nighttime refreezing (normally dampening cooling of the surface during the night) is the cause of deviations for the lower and upper ablation sites.

Section 3.1.2: Snow temperature

Snow temperature is a very good indicator of the performance of the model, and it is a very good idea to analyse and discuss this here. However, measuring snow temperature is in some sense trivial but using it is very complicated. With thermistor strings you can easily measure the snow temperature on a give location of the snow, either below or initially above the surface, depending how the string is installed. Given this installation, the temperature sensor moves with the snow pack in which is is burried, or stays at a given height above or below the measurement frame (like an Automated Weather Station) it is attached to. In all cases, the actual snow surface is moving away and towards the sensor all the time, so a sensor is never all the time at, say, 1 m depth below the surface. Compared to the data available online at the PROMICE website, the shown observational curve equals to those of "sensor 1" - although the online dataset has considerable datagaps in the summers of 2016 and 2018. The Fausto paper indeed state that this sensor is/was at 1 meter depth, but winter accumulation and summer snow melt are both well over 1 meter at TAS-A, so it could be - well, has been - anything like 0 to 2 meters. Given that sensor 2 has positive values in the summer of 2016 (after which the snow temperature sensors are reinstalled, visible in the shift in all readings), that happened for sensor 1 as well.

I really would like to see that the authors can retain this analysis, but that does require that they reconstruct the actual depth below the surface of "sensor 1" using the observed surface height - and not only for TAS A but for all stations used in Figure 5. After reconstructing the actual depth of the sensors, the adjacent model temperature can be extracted from the model data and compared to the observations.

If this is not possible, another way to assess the subsurface temperature is to replicate the evalation as provided by e.g. https://tc.copernicus.org/articles/15/1823/2021/.

And if this analysis is retained and improved, then it would be great if the results could be compared with subsurface temperatures from what is considered advanced surface models for either (or both) SEB models and or polar regional climate models, similar as requested for Figure 2.

L 287 & Figure 5: State explicitly that this Taylor diagram and bias on annual data.

Section 3.1.3 Trend in melting occurrences. Personally, I am more interested in the model performance than in the trend, as the latter has been documented in many papers already. So focus more on the statistics - in how many cases are the melting days well predicted, missed and are there many false alarms (and how does this compare to other models) - and remove the trendline.

Figure 8: please replace panels b and d by panels with the biases compared to the observations. A reader should be able to eyeball the improvement compared to CTL, but I don't want to eyeball how far off the GLA is from the observations, which is the current situation. And please be aware of the issues with the MODIS albedo if the solar zenith angle is larger than 70 degrees - which also may explain the positive bias in North Greenland in panel 8c.

L 338: I'm fine if you retain the reference to Zorzetto, but please be aware such parametrizations are already used for over a decade in polar adapted RCM like RACMO

and MAR, the latter through the snow model CROCUS. Acknowledge that with appropriate references.

Section 3.2.2. This analysis is useful, but a common practice in other papers is to compare modelled SMB against the in-situ observations compilation from Machguth (J Glac 62, 2016, currently being updated by DMI) like done in Noël et al, 2019, Sci Adv. (Supplementary figure 3) or a subset of that like van https://tc.copernicus.org/articles/15/1823/2021/.

L 367: rephrase, ice sheets are not glaciers.

L 373: Please explain briefly, in the methods section, the Kling-Gupta Efficiency. Now every reader not familiar to it, is forced to dive up Gupta et al.

Figure 11, panel a. Not the KGE is shown, but the difference or change in KGE. Adjust the caption accordingly.

Section 3.3: KGE and glaciers are not my expertise, but when KGE is a common measure to evaluate river discharge, there are other model evaluation studies that have given KGE scores - so cite a few other studies and their scores to give the reader a clue if the KGE scores GLA and CTL are good or poor.

---

## Author Comment (AC1)

Responses to Reviewer #1

The authors updated the parameterizations for glacier ice and snowpack within the land surface model and carried out validation over the Greenland Ice Sheet. The updated model demonstrated comparatively strong performance on the Greenland Ice Sheet and yielded valuable outcomes for the future development of ecLand. However, there remain several unclear aspects and areas lacking sufficient validation. I hope that the following comments will help improve the manuscript.

**Major comments:**

1. The differences between the existing model (CTL) and the updated model (GLA) are not clearly explained. For example, it is not clear what parameterizations the CTL used. Since this is a key part of the manuscript, I suggest showing the differences between the two models in a table to enhance clarity.

Following the recommendations from the reviewer we have amended the Methodology section to better describe the differences between CTL and GLA:

- To highlight the current parameterisations used for glacier points, we have modified Section 2.1 and added a new subsection, 2.1.1, named "Current treatment of glacier points in ecLand".
- Section 2.2 has been further divided in subsections to highlight the modified snow process that is discussed.
- A table has been added to better clarify the differences in each component of the snow scheme, see below:

**Table 1.** Summary of the differences in the representation of glacier grid points between the current model version (CTL) and the new glacier parameterisation (GLA).

| Parameter / Parameterisation | CTL | GLA |
|---|---|---|
| Sub-grid ice tile | None | Explicit ice tile with sub-grid fraction |
| Ice Thermodynamics | None | Included (4 layers) |
| Ice Albedo | None | Fixed, 0.4 |
| Ice Melting | None | Included (bare-ice exposure) |
| Snow Mass Balance | Fixed to 10 m SWE | Dynamic and capped to 10 m SWE, see Sect. 2.2.2 Snow Mass |
| Snow Albedo | Fixed, 0.82 | Dynamic, see Sect. 2.2.2 Snow Albedo |
| Snow Density | Fixed, 300 kg m$^{-3}$ | Dynamic, see Sect. 2.2.2 Snow Density |

2. The transition from validating against the Greenland Ice Sheet to evaluating river discharge in the Northern Hemisphere feels somewhat sudden. Validation using in-situ albedo may be challenging, but would it be possible to compare the land surface albedo in CTL or GLA with MODIS albedo across the Northern Hemisphere? At least, the authors should discuss whether the GLA experiment also shows a decrease in albedo outside of Greenland.

As suggested, we have examined the albedo differences between GLA and CTL (see Figures below). The first scatter plot compares grid-box average albedo for the boreal summer months over 2000–2019, considering only grid points with

glaciers/land ice (glacier mask > 0.1). The spatial map shows the composite albedo differences for the same period and season, providing context to the scatter plot.

Overall, at grid points where the glacier mask exceeds 0.5 the albedo is reduced. This reduction is due to the new dynamic albedo parameterization for snow over land ice in GLA, in contrast to the fixed value of 0.85 used in CTL. Things are more complex at grid points where the glacier mask is below 0.5: for grid points located in high altitude regions, the new sub-grid representation of ice leads to an increase in the grid-box average albedo; however coastal points show a reduction of albedo.

Because of the above, the associated increase in river discharge within the analysed basins is primarily linked to the exposure of bare ice, either fully resolved or sub-grid, which can melt during summer and thus contribute to runoff.

This discussion has been included in the revised manuscript, in Sect. 3.3, to interpret the changes in the river discharge as follows: "Changes in the river discharge may stem from a general reduction of snow albedo over glaciated surfaces in the Northern Hemisphere and the exposure of bare ice. For grid points located in high altitude regions and with sub-grid glaciers, there is a general increase in the grid-box average albedo in GLA compared to CTL (see Supplementary Figure Sx). Therefore, the increase in river discharge within the analysed basins is primarily linked to the exposure of bare ice, either fully resolved or sub-grid, which can melt during summer and thus contribute to runoff."

The plots have been added as supplementary material.

[Figure]

**Specific comments:**

Title: In my impression after reading this manuscript, the use of the term "global" feels somewhat excessive. Since the river discharge validation is conducted only for the Northern Hemisphere, it might be better to change the title to "local and regional impact," or remove the phrase ": local, regional and global impact."

> We have amended the title to better reflect the content of the manuscript as follows: "Enhancing the Representation of Glaciers and Ice Sheets in the ecLand Land-Surface Model: Impacts on Surface Energy Balance and Hydrology Across Scales."

L35 (Surface processes…): Add melting and refreezing processes.

Thanks, done.

L67: What do you mean by "physical capping"? Please add a clearer explanation.

Mottram et al. (2017) have introduced a hard limit to the surface temperature solver over ice and snow so that it does not exceed the melting point. This has been clarified in the revised version of the manuscript.

2 Methodology: It is better to briefly describe the difference between the GLA parameterization and the other LSM parameterization. Lee et al. (2024) might help you to compare GLA with LSMs (land surface models).

Thanks. This has been done as part of the revision of the "Methodology" section (see major comment).

L87: Does "fully coupled" mean "Atmosphere, land and Ocean coupling"?

The IFS runs by default as an atmosphere-land-ocean coupled model, but in the context of this paper we mean the coupling between the land-surface component, ecLand, and the atmospheric component ("IFS"). We have clarified this in the manuscript.

2.2 New glacier parameterisation: Although the GLA parameterizations are described, it is not clear what has changed compared to the CTL parameterizations. I would like to understand the differences between the two models before seeing results, so I suggest summarizing the characteristics of both models in a table.

We have addressed this suggestion by adding additional material to the Methodology Section.

L157-158: Previous studies have suggested that new snow density in the polar region exceeds 300 kg m-3 (Greuell and Konzelmann, 1994; Lenaerts et al., 2012; Niwano et al., 2018), so I have no objection to your assumption for the Greenland simulation. However, you should be careful if you apply the assumption to midlatitude areas. New snow density in midlatitudes is typically around 100 kg m-3 as indicated by previous studies (e.g., Niwano et al., 2012). Regarding this point, it should be stated whether changes in snow density parameterization affect snow albedo.

We thank the reviewer for this comment and acknowledge that using a limited dynamical range for the density of new snow over glaciers in the midlatitudes can be a limitation. However, we believe that this is still an improvement compared to the previous scheme, for which a resolved glacier point would have a constant snow density of 300 kg m-3, with no variability in time. We are currently working on spatialising the parameters of ecLand, to allow a more flexible use of parameter values across different climate conditions. This work will allow us to use different values of new snow density depending on the region and will be

evaluated in future work. We have included part of this discussion in the revised manuscript, see new Sect. 2.2.2 Snow Density.

L169-170: Please add the references for the values you used for Eq. (3). If the values are not based on previous studies, please add the rationale why you set the values.

The values were selected based on a preliminary parameter tuning experiment conducted prior to the analysis presented in the paper. In this experiment, the parameters were adjusted to achieve the best compromise between their impact on snow processes (as evaluated in this study) and on near-surface weather variables when the model is coupled to IFS for numerical weather predictions. We have clarified this aspect in the revised version of the manuscript, see new Sect. 2.2.2 Snow Albedo.

L192: If there is no alternative to the spatial SMB data other than RCM, it should be mentioned.

We use this product because it allows a comparison of the current scheme with state-of-the-art RCMs used to produce SMB estimates. A comparison with in situ observations would be valuable, for instance the observations compiled by Machguth et al. (2016). However, this would require a higher horizontal resolution, and consequently a more detailed glacier/ice-sheet mask, as most of the in situ observations compiled by Machguth are near the ice-sheet margin. In addition, the altitude difference between the observation location and the model grid point could further affect the SMB. We have clarified this in the revised manuscript, in Sect. 2.3, as follows: *"However, given that this product is based on RCMs, it will be used as a reference for comparison with current state-of-the-art models for SMB studies, rather than a validation dataset. A detailed comparison with in situ observations (see for instance Machguth et al., 2016) would require a high horizontal resolution and a more refined glacier/ice-sheet mask. Such analysis is beyond the scope of the present study, which is primarily focused on global 2D simulations in a close-to-operational setting (see Sect. 2.4.2)."*

L205 (different periods for each site): This is vague explanation. Please describe clearly. It would be helpful if you could make a table summarizing the experimental setting, including other experimental settings. A supplemental material might be good.

Thanks, we have added a table as supplemental material (Table S1) to summarise the experimental setting. The paragraph on the different periods has been reformulated as follows: *"The point scale simulations over the PROMICE stations described in Sect. 2.3 are run for different periods for each site, depending on the availability of observations at each location. To minimise spin-up effects, each site is simulated repeatedly over its available period until at least 30 years of spin-up are achieved, before performing the final simulation used for evaluation. Three types of experiments are run to evaluate the impact of the new glacier parameterisation depending on the forcing used to drive the model, as well as the glacier mask used to identify the glacier points, as summarised in Table S1."*

L220: Did you apply elevation correction to ERA5?

No, an elevation correction was not applied. This choice was made to better reflect the conditions of a coupled model run. While we acknowledge that applying such a correction could qualitatively affect the results (see also response to Reviewer #2 comment on "differences due to elevation difference"), it would not represent the specific conditions we intended to investigate. We have clarified this point in the revised manuscript, see Sect. 2.4.1.

2.4.2 2F global simulations: Please add an explanation regarding spinup simulations.

The model is initialised in 1970 in order to spin up thoroughly. Snow variables across the five layers are initialised using data from ERA5, following the "warm-start" procedure described in Arduini et al. 2019. The four-layer ice temperature is initialized based on the temperature of the lowest snow layer and is allowed to evolve dynamically. Given the relatively limited total thickness of the ice layers (10.86 m), they are expected to reach thermal equilibrium within approximately 20 years prior to the period used for evaluation (1990 onwards). We have included this discussion in the revised manuscript, see Sect.2.4.2.

Figure2 (skin temperature): "Skin temperature" and "surface temperature" are mixed up in the main text. If they mean the same thing, please unify either.

We have unified the terminology using surface temperature across the revised manuscript.

Figure2: What period does the analysis in this figure cover? Please add the period and season for the analysis.

Thanks, done.

Results: The Results section should be nominally limited to new results from the current observation or calculation and not include a literature review (L264, 281. 334...). I found that the authors' interpretations are included within this section (e.g. L280-284, L332-340). I suggest that you change "Results" section to "Results and Discussion" section.

We have modified the Section title to "Results and Discussion" following the reviewer's suggestion.

Line 242: Could you tell me about the specific scheme you improved?

We have amended the methodology section to better clarify the differences between GLA and CTL in order to provide a solid foundation for the reader throughout the manuscript.

Figure 3 (c): In the OBS experiment, temperature from PROMICE is used as an atmospheric forcing, yet panel (c) shows a bias of nearly 3°C during winter. The result looks strange. Please verify that there are no errors in the simulations or analyses. If no

errors are found, the bias may be due to ERA5 precipitation used in the OBS experiment. Additionally, I could not locate the CLIM experiment lines in the figure.

The land-surface component of the IFS, ecLand, generally struggles to simulate extremely cold wintertime surface temperatures over ice sheets. This issue has been documented in several studies over Antarctica (see for instance Dutra et al. 2015) and similarly applies to Greenland. The two contributing factors are: the excessive thermal inertia of the snowpack when vertical discretization is too coarse; excessive turbulent mixing in stably stratified conditions. We have added two supplementary figures illustrating the different components of the surface energy balance. In addition, the revised manuscript now includes a physical interpretation of the biases in Figure 3 and 4 (see Sect. 3.1.1).

Regarding the CLIM experiment in Figure 3c, this is underneath the ERA5 experiments, as those are equivalent for the accumulation sites. We have clarified this in the revised manuscript.

Figure 4 caption: Add the specific season you analyzed.

Thanks, done.

Line 270: Add the specific months.

Thanks, done.

Line 280: This paragraph is clearly a discussion.

Following the reviewer's suggestion we have renamed the section "Results and Discussion".

Figure 5: Please modify the legend. It looks like an old version. It is better to make the texts about the bias values larger.

Thanks, done.

Figure 6: I could not find the red dashed line at first. Please add that the red dashed line can be seen at 0 kg m-2.

Thanks, we have improved the quality of this figure following the reviewer's suggestions.

Line 310: It is better to add the explanation regarding snow layers in the models to the method section.

Following previous comments from the reviewer, we have moved the details on the discretisation to the Methodology section, focussing only on the discussion of the results in this paragraph.

Figure 8: As the color bar in the upper panels and the map in the bottom panels are close to each other, the labels on the color bar were misunderstood for the titles of the lower panels. In addition, please describe the difference between CTL and the validation dataset you analyzed to the labels of Figure 8a and c (for example, difference between Glacier minus CTL, like Figure 8b and d).

> This figure has been modified and improved following comments from Reviewer #1 and Reviewer #2.

Figure 11(a): The text for CTL and GLA is cluttered, so why not red and blue text for CTL and GLA, respectively?

> Thanks, we have modified the text accordingly to reviewer's suggestion.

Figure 11(b-e): It is hard to see each line. How about changing the line style to a dashed line for CTL and GLA?

> Thanks, done.

Line 393: Could the cause of this overestimation be the decrease in albedo in the GLA experiment? In the validation over the Greenland Ice Sheet, the GLA experiment showed a significant reduction in albedo. It is necessary to compare the land surface albedo from MODIS, CTL, and GLA across the Northern Hemisphere and discuss whether the new parameterization leads to a decrease in albedo over the Northern Hemisphere.

> This has been addressed as part of the reviewer's "Major comment" #2.

References:
https://www.ecmwf.int/sites/default/files/elibrary/2015/15262-understanding-ecmwf-winter-surface-temperature-biases-over-antarctica.pdf

---

## Author Comment (AC2)

Responses to Reviewer #2

The authors present results of evaluation tests of a new tile within the ecLAND module of the IFS-code of the ECMWF. I have read this paper with interest - the paper is generally well written - but needs improvement in the depth of the evaluation. The authors now convincingly show that the code update is an improvement. However, it would be good if the authors show also the performance against existing models, to see if the new tile/module performs similar as those models, or that there is still room for further improvement.

> This paper focuses on describing the improvements to ecLand, which is part of the IFS/ERA6 system, widely used as a global dataset. While an additional comparison to other models would be of interest, it is beyond the context of this manuscript. Our aim here is not to compare ecLand against other models, but rather to improve the representation of snow and glaciers within the coupled system for Numerical Weather Prediction (NWP) and reanalyses applications and therefore show an improvement to the overall model performance. We acknowledge that there remains room for improvement from a snow and ice perspective. However, because ecLand is part of an operational forecasting model, any significant changes to augment the realism of the snow/ice schemes may degrade NWP skills. For this reason, comparisons with other models, though potentially informative, would not add substantial value given the specific goals pursued here.

The essential improvements are those in the evaluation of the skin temperature, and especially of those of the subsurface temperature and to add an evaluation of in situ surface mass balance. I would genuinely like to see the technical improvements that I think are essential for better representation of the SEB and SMB without much coding - at the other hand I am afraid this request/wish will be in vain.

> As indicated above, the fact that ecLand is part of an operational model precludes such significant changes to the model at any one time. While we agree with the reviewer that the SEB and SMB could be improved, the impact on the NWP model, as well as future reanalyses would potentially be unforeseeable and must be carefully evaluated.

**Comments**

Introduction: glaciers and ice sheets are two different things, so by using "glaciers", ice sheets are sometimes forgotten. Therefore, I think it is better to replace "glaciers" of lines 24, 32, 39, 106 to "glaciers and ice sheets", "glaciers" of lines 30, 94, 97, 100, 107 to "glaciated surfaces". L 108 is a special case, I leave it to the authors to make it more general. And check other instances when I have missed the word "glacier" in this list.

> Thanks. We have amended the text throughout to make this aspect more precise.

L 34: Greenland Ice Sheet - so with capitals.

    Thanks, done.

L 38: Don't forget that for ice sheet, Earth System Models (like CESM2, UKESM) perform increasingly well (if the ESM developers try to model the SMB well), so regional climate models are not the only alternative of using observations only. These things are discussed later in the introduction too, so this formulation is already somewhat inconsequent.

    Thanks, we have amended the text to better reflect this aspect.

L 43-54: I don't think this sidestep to ice dynamics is relevant for this manuscript nor introduction. For example, the initial-condition problem of ice sheet models is completely different to those of atmospheric models. I propose to remove this paragraph.

    In hindsight, we agree with the reviewer's comment and have removed this paragraph in the revised manuscript.

L 48: I think this reference if outdated, take a newer one if you decide to retain this paragraph.

    This paragraph has been removed following reviewer's previous suggestion.

L 65: A regional climate model is not an Earth System model. So, HARMONIE-AROME improvement are not an example of ESM improvements.

    Thanks, we have rephrased this sentence as follows: *"Efforts to improve the representation of ice sheets and glaciers in NWP models have been relatively limited compared to the advances in climate models. Mottram et al. (2017) improved the representation of melting events in the HARMONIE-AROME regional model for NWP applications, by including an upper threshold on the surface temperature of the ice surface (i.e. melting point), using the remaining energy to melt the snowpack."*

L 74-82: Given that the land-ice parameterization of ecLand is simple and uses very few layers compared to the advanced schemes in, particularly, polar adapted Regional Climate Models, I would like that the authors formulate very specifically what their new parameterization should do well, and what not necessarily. For example, is that the SEB (including albedo evolution) for the typical range of a weather forecast (14-21 days), on seasonal timescales; does it include the surface mass balance (including refreezing effects)?

    We thank the reviewer for this comment. We have clarified the scope and purpose of the new scheme in the introduction. Also, we have added a table in the Methodology section to highlight the main changes and process improvements in

the new scheme compared to the old one (see response to Reviewer #1 main comment).

L 94: As far as I can recall, this simple snow scheme has been used also in earlier IFS cycle. If so, replace by "In CY49R1 (the NWP ....2024) and preceding versions since version <first version with this code>, glaciers and ice sheets are ...."

> Glaciated surfaces have been simulated using the same formulation for many cycles (decades, possibly) of the IFS. We have clarified this sentence adding "preceding versions" as suggested by the reviewer.

L 135: This method of confining the skin temperature to the melting point is already long in the IFS code even before Arduini 2019. Still, I remain to the opinion that this workaround is a poor solution, given that one can easily solve this issue in a mathematically sound and numerical simple way. We have used previous IFS versions (e.g. CY33R1) and in those versions considerable errors arose. Specifically, the time step after observing melt, the initial skin temperature guess was below the freezing point, the large conductivity number was not used, and the actual skin temperature became again well above the melting point. Even if this problem does not arise in this version, the method of Arduini leads to inconsistent skin temperatures if the uppermost snow or ice layer is not at the melting point.

Our solution was and is the following: When one observes that the skin temperature of a snow/ice/glacier tile is above the melting point, we keep the skin conductivity as is, but (simply) apply that one thus knows that the skin temperatures of that tile is at the melting point. If those tiles cover the whole grid box, one only need recalculate the fluxes. If these melting surface cover only a part of the grid box, we solve the SEB again, applying that we know the skin temperature of these melting snow/ice tiles, while the other tiles remain unknown as in the normal linearized SEB solve method. The authors can have more extensive documentation of our approach if they wish to have.

> What the reviewer is referring to in the first part of their comment is a different method, not the one described in Arduini et al. 2019. As far as we know, the method has been documented and tested in coupled simulations in Arduini et al 2019, but we are happy to include any other reference, pointing to this specific implementation for offline and coupled simulations, that we are not aware of. The method described in the second part of their comment is very similar to what we have implemented in IFS, with the difference that the surface temperature over snow/ice tiles is constrained by setting the temperature of the underlying surface to the melting point and using a large skin conductivity. For clarity, the reviewer can have a look at the offline implementation in ecLand, around the LREPEAT switch: https://github.com/ecmwf-ifs/ecland/blob/main/src/surf/offline/driver/vdfdifh1s.F90
>
> More generally, when indicating an IFS cycle or reference, we refer to the cycle in which codes get operational and used for Numerical Weather Prediction, rather

than research branches in which code was added, but not active, under a logical switch.

Section2.2.2:
To me, this decision to not create separate prognostic variables for glaciated variables is a very, very poor compromise. For the glacial ice it is acceptable - sea ice is not active over land - but for snow I really do not see the urgency to mess up your snow physics to safe 4x4 prognostic variables (layer mass/thickness, layer density, layer temperature, layer water content). Snow surfaces, and particularly the snow albedo, is a classic example of a non-linear process, so the aggregate of two implies that now both are wrong.
We have no longer the computers of the 70s for which fast memory was a severe limiting factor and is the fancy type system of IFS not specifically set up to allow for adding new variables without having to adjust the code from top-to-bottom? It sounds like that the authors were allowed to play around and improve the representation of glaciated surfaces; under the condition they won't bother the rest of the ECMWF-IFS community in any way. Don't get me wrong, I applaud the efforts of the authors to improve the representation of glaciated surfaces, but this compromise is very typical of the general and decades-long neglect of glaciers and ice sheet surfaces by the ECMWF.

Ideally, this poor compromise is rectified, and glacial snow is separated from land snow (and if you are doing that, please also separate snow over low vegetation and the snow below high vegetation into two independent snow layers sets, as that mix-up is equally bad). But I also do understand this strong suggestion (separate variables) is infeasible to effectuate (now), and I am aware this rant leads to zero change on the code or paper. However, hopefully it encourages the authors (or their successors) to fight even harder for a proper representation of glaciated surfaces in the IFC code. In all cases, I would like to see a longer motivation in the rebuttal why this poor compromise has been taken.

> We thank the reviewer for this detailed and interesting comment. The decision of not including additional prognostic variables was motivated by the fact that the scientific improvements were foreseen limited, relative to the substantial technical effort required to implement and validate them correctly. We expand on these two elements in the following.
>
> Firstly, in response to the comment that "the aggregate of two implies that now both are wrong", we acknowledge that having two different prognostic variables is, in principle, the best solution. However, in the present work we are looking at the difference with the current model formulation. In this respect, the evaluation of albedo did not give any indication of degradation for the heterogeneous grid-points around the coast of Greenland. The evaluation of river discharge, an integrated measure of snowmelt over a river basin, was also showing improvements in most of the considered basins in the Northern Hemisphere. The latter is quite significant, as an erroneous snow albedo, both over "land" and land-ice tiles, would very likely lead to an erroneous snowmelt and therefore a degradation in skills in the simulation of the river discharge.

Secondly, although weather forecasting is not the primary focus of this study, it is central to ECMWF applications. From that perspective, the new scheme yields approximately an 8% improvement in 2-metre temperature forecasts errors around the Antarctic coast during summer, suggesting an enhanced representation of the surface energy balance. In addition to that, for weather prediction applications we believe that model complexity should always be balanced between the different earth system components. Therefore, it is not immediate that additional complexity, like five additional prognostic variables, would lead to better results.

We agree with the reviewer that computational resources have advanced significantly since the 1970s, and memory limitations are now a less critical constraint. However, the complexity of modern code bases has also increased substantially, such that introducing a new component requires many changes in several components, from initialisation to data assimilation and archiving, to avoid undesired effects. From a technical and scientific perspective there are several challenges that should be considered that we believe require dedicated future work:

- in an operational system for NWP, the additional prognostic fields should be correctly initialised to avoid spin-up issues in the subsequent forecast or in a hindcast setup to calibrate seasonal and sub-seasonal predictions.
- Usage of additional prognostic variables for snow tiles should be evaluated in the 4D Variational Assimilation (4D-Var) system, in particular for the observation operators making use of the snow variables (e.g. microwave observation operators).
- The new variables would require additional GRIB2 codes that must be proposed to WMO and approved, before the fields can be disseminated.

We can reassure the reviewer that our intention is to push the boundaries of snow/glacier modelling within the IFS. However, as within all operational centres, such changes need to be implemented gradually.

L 168: Please specify what Tsn is. It could be the temperature of the uppermost snow layer or the glaciated surface skin temperature - I don't know now. I presume the authors are aware that this snow albedo (Eq. 3) includes two major simplifications. Firstly, the snow albedo is not a function of temperature (or density-per sé), but of snow grain size (and to some extend to the solar zenith angle). Secondly, the snow albedo is very strongly dependent on the wavelength, almost always 1 for UV to yellow; strongly varying for "red" and zero-ish for near-infra red - see e.g. Gardner and Sharp, GRL, 2010, or Van Dalum et al, The Cryosphere, 2020). So, using one albedo value number is similarly bad as running ecRAD with only one G-band. Again, I know it is not realistic to expect the authors to use a state-of-the-art grain-size based snow albedo scheme, but it is good to mention in the manuscript that this albedo is not regarded as

state-of-the-art and specify its limitations with a reference to Gardner and Sharp, GRL, 2010 or a similar paper.

> We thank the reviewer for this comment. We have amended the text to clarify the limitations of the implemented snow albedo scheme with the relevant references. The update of the snow albedo scheme is in our future plans and will require dedicated work. Following what we have said in the previous response, we believe that complexity *per se* might be detrimental in a coupled model due to compensating biases between different processes. Therefore, implementation of more complex schemes based on snow grain size (or a proxy of it) and using several spectral bands will require a careful implementation and retuning of other model components.

Section 2.4: It doesn't become completely clear to me how the experiments are carried out. This holds for both the point-scale as 2D global simulations. I understood from the description that the land model is rerun, but the atmospheric model not. Whether I got it right or not, explain in more detail which parts of the code have been rerun and which not. Furthermore, please specify which fields/fluxes are updated/adjusted in the experiments, and which fields/fluxes were kept constant. I conclude from the paper that the SEB has been recomputed, but I don't understand how that is done as that is far from trivial to do afterwards online within the IFS framework (e.g. one needs the derivatives of fluxes to close the SEB and I would be seriously surprised as these derivatives are available from the ERA5 simulation.).

> We thank the reviewer for this comment. The land component of the IFS, ecLand, is fully externalised and can be run "offline" forced with atmospheric fluxes and state variables as boundary conditions with a certain forcing frequency. In this configuration, the forcing variables are liquid and solid precipitation, downwelling longwave and shortwave radiation; wind speed, temperature and specific humidity at the lowest model level; surface pressure. The forcing variables are kept constant if the forcing frequency is greater than the model's time step. For these experiments, the forcing frequency is 1 hour, and the model's time step is 30 minutes. With these inputs the SEB is solved using the implicit method described in Best et al. (2004), providing the required fluxes to run the land-surface components. The caveat, as in all land-surface only simulations, is that there is no feedback to the atmosphere due to changes in the land variables (e.g. snow). We have amended the text adding more details on the simulation setup in Sect. 2.4 and summarised experimental settings in Table S1 (as supplementary material).

Figure 1 & 8: please use a map projection that shows Greenland with the right width to length ratio, so either Lambertian, Polar Stereographic or rotated lat-lon.

> Thanks, done.

L 175: From the PROMICE dataset, the authors use the skin temperature. However, this temperature is not measured. I presume that the authors use the temperature derived from the upwelling longwave radiation. Please specify that explicitly here.

> Thanks, the temperature is derived from the upwelling longwave radiation. We have amended the text to clarify this aspect.

L 220: Make this description of the CLIM data more specific. Fgl is thus not 1 - otherwise that would have been stated - but still it would be sensible to use the tiled skin temperature for Figure 2-4. State which values have been used - grid-box average skin temperature or tile skin temperature. From line 275 I conclude that grid box averaged skin temperatures are used, it is better to adjust this - as the paper is about evaluating the glacier tile and the observations are on the ice sheet. So, why is not the tiled skin temperature analyzed? This parameter can be exported, so practical reasons are not impeding this.

> We guess the reviewer is referring to lines 223-225, describing the "E5-CLIM" experiments, not line 220 describing the "E5" experiments. Regarding the "E5-CLIM" experiments, the purpose of the experiment was to evaluate the schemes in a sort of "operational" setting when the global model is run using a fractional glacier mask. With this purpose, it makes sense to evaluate the grid-box average skin temperature, as this is what would be passed to the atmospheric model in coupled simulations. We have clarified this aspect in the revised version of the manuscript.

I like the analysis in Figures 2-4 - it is a good method to show what the new module can and cannot. The drawback is that it is hard to compare how well this new module performs compared to existing glacier surface descriptions, as I haven't seen it in other papers. Therefore, assuming that the PROMICE skin temperature is derived from the upwelling longwave radiation, the golden standard is the modelled skin temperature with a SEB model (e.g. https://doi.org/10.1017/jog.2024.68 ). It would be good add the performance of such a dataset, to compare it against the CTL-OBS and GLA-OBS results. Similarly, the "E5" and "E5-CLIM" could be compared of the performance of a polar adapted RCM like MAR (https://doi.org/10.5194/tc-14-957-2020) or RACMO (https://doi.org/10.5194/tc-12-811-2018 - although this dataset is outdated). I am quite sure the required data for such an analysis is available for the authors to be used.

> We thank the reviewer for this comment and the suggested references. Figures 2 to 4 already compares the surface temperature resulting from the SEB to the one diagnosed from the observed upwelling longwave radiation. We agree that a comparison with other models could provide additional insights for the reader. However, such an analysis lies beyond the scope of the present study, which is specifically focused on evaluating the new module against the current implementation and showing the improvement for a globally used model. A fair and meaningful comparison with a Regional Climate Model, as suggested, would require a dedicated experimental design and a separate study, which would be interesting to consider in the future.

L 252-260: The interpretation of Figure 3 is complicated without analysing the SEB and the T2m. An T2m temperature analysis could indicate if the atmospheric conditions of the E5 and E5-CLIM simulations are colder/warmer than those in the observations. Similarly the SEB analysis (against the PROMICE data) could indicate why the conditions are colder than observed. These figures don't need to be added to the manuscript, but such an analysis add depth to paragraph, which is now not much more than "Hmmm, our model is warmer/colder than observed." In that respect, the 3K warm winter bias of both OBS is remarkable, as for this experiment the 2m is "correct". In short, figure out why the new and old model deviate, and report that in the revised manuscript.

We thank the reviewer for this comment. We have looked at the Surface energy balance components to diagnose how the other energy fluxes respond to the changes in the ice and snow parameterisations. With regards to Figure 3, the differences in the wintertime bias between the OBS and ERA5 experiments are mainly attributable to downwelling longwave radiation ("LWdown"), which is systematically higher in the observations than in ERA5 (see below for the accumulation sites). The weaker LWdown forcing in ERA5 leads to lower surface temperatures in the E5 experiments, which in turn explains why CTL-ERA5 and GLA-ERA5 exhibit a smaller bias compared to CTL-OBS and GLA-OBS. Importantly, lower surface temperatures increase the surface–air temperature gradient, thereby enhancing the sensible heat flux ("Qh") in the ERA5 experiments. It is well known in the literature that turbulent mixing tends to be overestimated in ecLand/IFS, which helps to explain why the E5 experiments still display a positive wintertime bias despite the underestimated LWdown relative to the observations.

We have included this discussion in the revised manuscript, see Sect. 3.1.1 and have amended the text to provide more physical understanding on the causes of the differences in Figure 3 and 4 between CTL and GLA; we have added the Figures of the mean annual climatology and diurnal cycle of fluxes as supplementary material of the revised manuscript.

[Figure]

Figure 4: add in the caption that summer months are evaluated.

Thanks, we have clarified that June-July-August (JJA) are evaluated.

Concerning Figure 3-4, to which extend are the difference due to elevation difference between the observational site and the height of the grid box? Elevation biases induce temperature biases which have nothing to do with poor functioning parameterizations. Remove the effect of an elevation bias (if present).

We have computed the lapse-rate correction at each site to separate errors due to elevation differences from those due to forcing uncertainties. The average lapse rate for the "low", "upper" and "accumulation" sites is 0.29 °C, 0.15 °C, -0.13 °C, respectively. For this reason, we have chosen not to modify the plots, but we now explicitly mention these results in the revised text for clarity.

Concerning Figure 4, again I would like to challenge authors to dive a bit more deeper into the 'why' the model is deviating from the observations. Very little physical explanations are given, and the manuscript (and your understanding of the model performance) will be improved if this indepth analysis is made. Again, the avenue to get this insight is through analysing the SEB, and again these figures don't have to be added to the manuscript (possibly supplementary materials), but allows for a more physical explanation why the model is deviating. I would guess that the underestimated cycle for the high locations is due to too high effective thermal capacity of the snow layer (being 50 cm thick), while alternatively missed nighttime refreezing (normally dampening cooling of the surface during the night) is the cause of deviations for the lower and upper ablation sites.

We have responded to this point in the previous comment regarding "L 252-260". As responded in that comment, we have included a discussion on the changes to the surface energy fluxes to have a better physical understanding of the differences in the revised manuscript in Sect. 3.1.1.

Section 3.1.2: Snow temperature

Snow temperature is a very good indicator of the performance of the model, and it is a very good idea to analyse and discuss this here. However, measuring snow temperature is in some sense trivial but using it is very complicated. With thermistor strings you can easily measure the snow temperature on a give location of the snow, either below or initially above the surface, depending how the string is installed. Given this installation, the temperature sensor moves with the snow pack in which is is burried, or stays at a given height above or below the measurement frame (like an Automated Weather Station) it is attached to. In all cases, the actual snow surface is moving away and towards the sensor all the time, so a sensor is never all the time at, say, 1 m depth below the surface. Compared to the data available online at the PROMICE website, the shown observational curve equals to those of "sensor 1" - although the online dataset has considerable datagaps in the summers of 2016 and 2018. The Fausto paper indeed state that this sensor is/was at 1 meter depth, but winter accumulation and summer snow melt are both well over 1 meter at TAS-A, so it could be - well, has been - anything like 0 to 2 meters. Given that sensor 2 has positive values in the summer of 2016 (after which the snow temperature sensors are reinstalled, visible in the shift in all readings), that happened for sensor 1 as well.

I really would like to see that the authors can retain this analysis, but that does require that they reconstruct the actual depth below the surface of "sensor 1" using the observed surface height - and not only for TAS A but for all stations used in Figure 5. After reconstructing the actual depth of the sensors, the adjacent model temperature can be extracted from the model data and compared to the observations.

If this is not possible, another way to assess the subsurface temperature is to replicate the evalution as provided by e.g. https://tc.copernicus.org/articles/15/1823/2021/.

And if this analysis is retained and improved, then it would be great if the results could be compared with subsurface temperatures from what is considered advanced surface models for either (or both) SEB models and or polar regional climate models, similar as requested for Figure 2.

> We have followed the suggestion and used the sensor height measurements relative to the snow surface to diagnose the actual depth of the temperature sensors. This depth can vary substantially due to snow accumulation, reaching up to 3 m during the winter months. Accounting for this variability significantly improves the agreement of the GLA experiment with the observations (see below). Previously, the model tended to be too cold in winter because it considered temperatures from a snow layer closer to the surface, which is more strongly influenced by the colder atmosphere above. We have included the new figure in the revised manuscript (new Fig. 6, see below) and updated the text accordingly in Sect. 3.1.2. Even though interesting, we believe the comparison with SEB models regional climate models is beyond the scope of the current work and should be explored in future work.

[Figure]

L 287 & Figure 5: State explicitly that this Taylor diagram and bias on annual data.

Thanks, done.

Section 3.1.3 Trend in melting occurrences. Personally, I am more interested in the model performance than in the trend, as the latter has been documented in many papers already. So focus more on the statistics - in how many cases are the melting days well predicted, missed and are there many false alarms (and how does this compare to other models) - and remove the trendline.

A statistical comparison using a skill score (e.g., the threat score) is already presented in Figure 9 of the original manuscript. The purpose of Figure 7 is to evaluate how the current and new modules reproduce trends in melting occurrences, which is an important feature to assess for a model which is intended to be used for reanalysis products (e.g. future ERA-Land datasets). We have revised the text in Sect. 3.1.3 to clarify this scope.

Figure 8: please replace panels b and d by panels with the biases compared to the observations. A reader should be able to eyeball the improvement compared to CTL, but I don't want to eyeball how far off the GLA is from the observations, which is the current situation. And please be aware of the issues with the MODIS albedo if the solar zenith angle is larger than 70 degrees - which also may explain the positive bias in North Greenland in panel 8c.

The purpose of our analysis is to highlight the improvement or degradation of the new model relative to the control experiment. For this reason, we consider it more appropriate to present the absolute bias difference, as this illustrates the regions where the new scheme improves or degrades albedo and melting occurrences. We also appreciate the comment regarding MODIS albedo uncertainty and have now included this point in the manuscript.

L 338: I'm fine if you retain the reference to Zorzetto, but please be aware such parametrizations are already used for over a decade in polar adapted RCM like RACMO

and MAR, the latter through the snow model CROCUS. Acknowledge that with appropriate references.

> We have added the recent reference to Zorzetto as it was a recent example of a global model using a more physical and complex albedo parameterisation. For more completeness, we have added a reference to CROCUS in the revised manuscript.

Section 3.2.2. This analysis is useful, but a common practice in other papers is to compare modelled SMB against the in-situ observations compilation from Machguth (J Glac 62, 2016, currently being updated by DMI) like done in Noël et al, 2019, Sci Adv. (Supplementary figure 3) or a subset of that like van https://tc.copernicus.org/articles/15/1823/2021/.

> We agree with the reviewer that a comparison with in situ observations of SMB would be valuable. However, this would require a higher horizontal resolution, and consequently a more detailed glacier/ice-sheet mask, as most of the in situ observations compiled by Machguth are near the ice-sheet margin. In addition, the altitude difference between the observation location and the model grid point could further affect the SMB. A preliminary analysis with a subset of the observations compiled by Machguth and used by van Dalum et al. (2021) shows qualitatively similar results to Figure 10b in the manuscript: the SMB in the CTL experiment being always positive (i.e. no mass loss), whereas the SMB in the GLA experiment shows better agreement with the observed values (see below). As the manuscript already contains several figures and comparison with various datasets, we chose not to include this additional analysis here but will consider it as part of future work.

[Figure]

L 367: rephrase, ice sheets are not glaciers.

Thanks, done.

L 373: Please explain briefly, in the methods section, the Kling-Gupta Efficiency. Now every reader not familiar to it, is forced to dive up Gupta et al.

We have amended the text including more details on the Kling-Gupta Efficiency and its common usage in hydrology.

Figure 11, panel a. Not the KGE is shown, but the difference or change in KGE. Adjust the caption accordingly.

Thanks, done.

Section 3.3: KGE and glaciers are not my expertise, but when KGE is a common measure to evaluate river discharge, there are other model evaluation studies that have given KGE scores - so cite a few other studies and their scores to give the reader a clue if the KGE scores GLA and CTL are good or poor.

A common strategy is to compare the performance of the model with the observed mean flow. Knoben et al. (2019) have shown that a KGE >~ -0.41 indicates that a model improves on the mean flow benchmark. This corresponds to a Nash-Suttcliffe Efficiency (NSE) value of zero. However, in this study we are more interested in the difference of the KGE between CTL and GLA to highlight if the new parameterisation improves or degrades the hydrological skills of the current modelling system. We have included the reference value of -0.41 in the revised manuscript, when discussing the KGE, as well as the KGE values of GLA and CTL in Figure 11 among with the values of the components of the KGE.

**References**

Best, Martin J., et al. "A proposed structure for coupling tiled surfaces with the planetary boundary layer." *Journal of hydrometeorology* 5.6 (2004): 1271-1278.

Knoben, W. J. M., Freer, J. E., and Woods, R. A.: Technical note: Inherent benchmark or not? Comparing Nash–Sutcliffe and Kling–Gupta efficiency scores, Hydrol. Earth Syst. Sci., 23, 4323–4331, https://doi.org/10.5194/hess-23-4323-2019, 2019.

---

## Author Response (AR2)

**Responses to Reviewer #1, Minor revisions**

*We thank the reviewer for their additional feedback on the revised manuscript. In the following, Reviewer's comments are shown in black, whereas the Authors' responses are shown in light blue.*

**Specific comments:**

L60: Since this is the first appearance of the term NWP, please spell it out in full.

Done.

L80: The word Regional may be unnecessary. As shown in the table in the appendix, the experiments appear to have been conducted either at the point scale or at the global scale.

Done, removed the word "Regional".

L172: Please clarify what type of snow has a density of 315 kg m−3. It would be helpful to provide an explanation similar to that for 280 kg m−3 (if possible).

We have added the following text to clarify what type of snow has a density of 315 kg m-3: *"(for relatively older snow)"*

L193: "aging".

Done.

L204: Please specify the value used as emissivity.

This sentence has been modified as follows: "The surface temperature is derived from the upwelling longwave radiation assuming a constant emissivity of 0.97 (Fausto et al., 2021)."

L265: The experiment name "CLIM" is somewhat unclear. I recommend changing it to a name that indicates the difference from the "E5" experiment—perhaps something reflecting the glacier mask.

The setup of the different experiments is reported in Sect. 2.4.1. We consider changing the experiment identifier at this stage potentially risky, as it could introduce inconsistencies throughout the manuscript. For this reason, we have retained the original identifier. To improve clarity, we have added the following note to the caption of Table S1: *"See Sect. 2.4.1 for a detailed description of the experimental setups."*

L308–309: The sentence "The average lapse rate for ..." is difficult to understand. Why is the unit given in °C when referring to a lapse rate?

We have modified this sentence to clarify its meaning: *"The average lapse rate **correction** for the low, upper and accumulation sites is 0.29 °C, 0.15 °C, -0.13 °C, respectively. "*

Figure 1: The purple color is hard to distinguish from red or blue. Please consider using a different color for clarity.

Done. We have changed the purple text to black.

Table 1: The entry "None" for the ice physical properties in CTL is not sufficiently descriptive. Please clarify what these mean. In CTL, were the physical properties of bare ice assumed to be identical to those of snow after the snow layer disappeared? It would also be informative to indicate the CTL model version and corresponding references in the table. Moreover, please include the layer thicknesses used for the Ice thermodynamics in GLA.

We have amended Table 1 following the reviewer suggestions (see below). We have replaced the entry "None" with a more descriptive text; included the model version into the Table caption; include the layer thicknesses in the "Ice Thermodynamics" entry of the Table. We think adding references in the table would make this table confusing to read and a reader can find the relevant references into the main text.

**Table 1.** Summary of the glacier and ice sheets processes represented in the current model version (CTL, ecLand CY49R1) and the new "glacier" parameterisation (GLA, ecLand CY50R1). See Sect. 2 for more details.

| Parameter / Parameterisation | CTL | GLA |
|---|---|---|
| Sub-grid ice tile | No explicit ice tile; dominant ice points prescribed with 10 m SWE | Explicit ice tile with sub-grid fraction |
| Ice Thermodynamics | Not considered, assuming only snow with no underlying ice | 4-layers ice scheme with layer thicknesses of 0.07 m, 0.21 m, 0.72 m and 9.86 m |
| Ice Albedo | Not considered, assuming only snow with no underlying ice | Fixed, set to 0.4 |
| Ice Melting | Not considered | Included (bare-ice exposure) |
| Snow Mass Balance | Fixed to 10 m SWE | Dynamic and capped to 10 m SWE, see Sect. 2.2.2 Snow Mass |
| Snow Albedo | Fixed, 0.82 | Dynamic, see Sect. 2.2.2 Snow Albedo |
| Snow Density | Fixed, 300 kg m$^{-3}$ | Dynamic, see Sect. 2.2.2 Snow Density |
| Snow liquid water | Not allowed | Allowed, with percolation and refreezing |

Figure 6: The legend item lwcs could be mistaken for a new experiment name, and obs is also potentially confusing. Consider renaming these to indicate both the variable and the dataset (e.g., Temp. (GLA-OBS) or Lwcs (GLA-OBS)).

Done.

Figure 8: It would be clearer if panels (b) and (d) were expressed as Glacier minus Obs. Currently, panels (a) and (b) show CTL minus OBS, whereas panels (c) and (d) show Glacier minus CTL, reversing the sign convention for CTL and potentially confusing readers.

The panels (b) and (d) show the difference in the absolute biases of GLA and CTL, against the Observations. This represents the change in the magnitude of the bias, with negative values indicating a reduction and positive values indicating an increase in the bias magnitude. We have clarified the labels of panels (b) and (d) to better reflect this point.

Supplementary Material: To distinguish supplementary tables and figures from those in the main text, Table 1 should be labeled as Table S1 (and other supplementary materials renamed accordingly).

Done.